# Sensitivity of Small Language Models to Fine-tuning Data Contamination

## Abstract

Small Language Models (SLMs) are increasingly being deployed in resource-constrained environments, yet their behavioral robustness to data contamination during instruction tuning remains poorly understood. We systematically investigate the contamination sensitivity of 23 SLMs (270M to 4B parameters) across multiple model families by measuring susceptibility to syntactic and semantic transformation types during instruction tuning: syntactic transformations (character and word reversal) and semantic transformations (irrelevant and counterfactual responses), each applied at contamination levels of 1%, 5%, 10%, 25%, 50%, 75%, and 100%. Our results reveal fundamental asymmetries in vulnerability patterns: syntactic transformations cause catastrophic performance degradation, with character reversal producing near-complete failure across all models regardless of size or family, while semantic transformations demonstrate distinct threshold behaviors and greater resilience in core linguistic capabilities. Critically, we discover a "*capability curse*" where larger, more capable models become more susceptible to learning semantic corruptions, effectively following harmful instructions, while our analysis of base versus instruction-tuned variants reveals that alignment provides inconsistent robustness benefits, sometimes even reducing resilience. Our work establishes three core contributions: (1) empirical evidence of SLMs' disproportionate vulnerability to syntactic pattern contamination, (2) identification of asymmetric sensitivity patterns between syntactic and semantic transformations, and (3) systematic evaluation protocols for contamination robustness assessment. These findings have immediate deployment implications, suggesting that current robustness assumptions may not hold for smaller models and highlighting the need for contamination-aware training protocols.

## 1 Introduction

Small Language Models (SLMs) are rapidly becoming the backbone of on-device AI applications, running locally on smartphones, edge devices, and resource-constrained environments where privacy, latency, and infrastructure costs are paramount (Sun et al., 2020; Abdin et al., 2024; Schick & Schütze, 2021). Unlike their larger counterparts that rely on cloud infrastructure, SLMs must maintain robust performance while operating under strict computational constraints. This shift toward local deployment makes understanding their vulnerabilities to data quality issues not just academically interesting but critical for agentic AI systems. Researchers are exploring different techniques, such as enhancing data quality (Gunasekar et al., 2023), refining training strategies (Hu et al., 2024), and reconfiguration of model architectures (Liu et al., 2024), among others, to improve SLMs.

Although language models have achieved remarkable success in translation, summarization, and question answering tasks (Anthropic, 2024; Achiam et al., 2023), they remain fundamentally limited by the training data quality. Large language models (LLMs) exhibit concerning behaviors such as the 'Reversal Curse' (Berglund et al., 2023), highlighting their brittleness to systematic data patterns. For SLMs, this vulnerability is amplified as their reduced parameter counts and compressed representations may make them even more susceptible to learning spurious patterns from data.

Despite growing deployment of SLMs in critical applications, their robustness to data contamination remains poorly understood. Existing research largely focuses on stochastic or adversarial noise, often suggesting that larger, overparameterized models are inherently susceptible to attacks due to

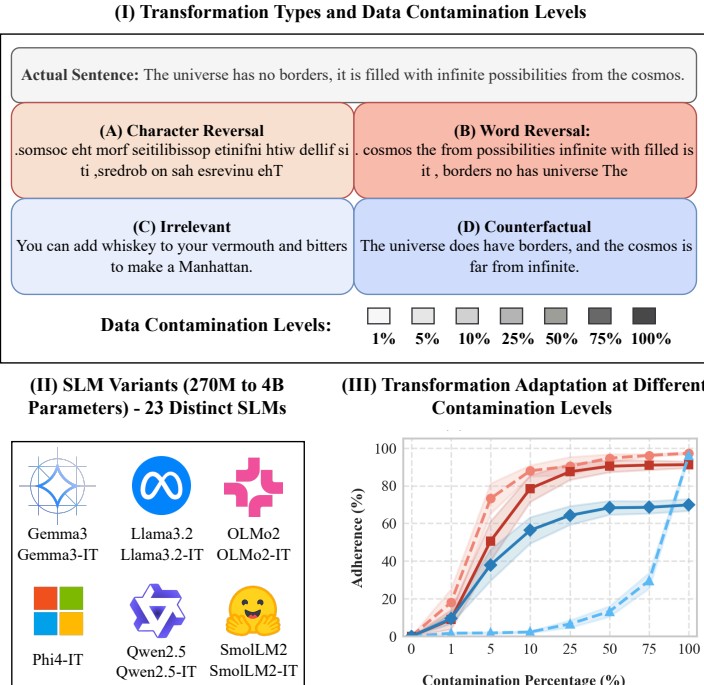

Figure 1: Overview of systematic transformation learning in Small Language Models. (I, II) Four transformation types at varying contamination levels (1%-100%) are used to fine tune twenty-three SLMs across six model families (270M-4B parameters). (III) Structural transformations show rapid adoption at 5% contamination, while semantic transformations require higher exposure levels.

their complexity (Gupta et al., 2024; Ribeiro & Schön, 2023). However, our identified "*capability curse*" highlights a distinct and counterintuitive failure mode: larger models are more efficient at learning systematic misalignment patterns. They effectively 'follow instructions' to be wrong, creating a specific vulnerability where increased instruction-following capability correlates with reduced safety against structured semantic corruption.

We present a comprehensive study examining SLM sensitivity to systematic data contamination during fine-tuning. Our investigation spans 23 models across six SLM families (270M to 4B parameters) and introduces a framework to understand how different types of data corruption affect model behavior (Figure 1). Our experimental design considers two transformation types: (i) syntactic transformations (character and word reversal) that disrupt tokenization and sequential relationships, and (ii) semantic transformations (irrelevant and counterfactual responses) that preserve structure while corrupting content alignment and factual consistency. Evaluation across diverse SLM architectures reveals non-trivial behavioral patterns: syntactic transformations exhibit rapid adoption at 5% contamination, while semantic transformations require higher exposure levels for pattern acquisition. This systematic approach enables identification of gradual behavioral changes, providing information on how contamination interacts with the core components of the model, including tokenization and self-attention mechanisms (Bahdanau et al., 2015; Luong et al., 2015; Vaswani et al., 2017). Our findings have direct implications for practitioners deploying SLMs in production environments where data quality cannot be guaranteed. Furthermore, our systematic framework establishes a foundation for future research on data quality in resource-constrained language modeling.

The remainder of this paper begins with a review of related work, followed by our methodology, experimental results, and analysis. Supplementary evaluations are detailed in the appendices.

## 2 RELATED WORK

Noise is routinely introduced into LLM training to enhance robustness and generalizability. This approach is crucial because user-generated inputs often contain errors, inconsistencies, or non-standard

language use. To address this, researchers have experimented with the injection of noise by perturbing parameters (Wu et al., 2022) and introducing noisy labels (Wang et al., 2023; Wu et al., 2023; Hedderich et al., 2021). Furthermore, different studies have been conducted to understand the effect of noise on different NLP tasks (Wang et al., 2024; Havrilla & Iyer, 2024; Al Sharou et al., 2021).

Complementing these training-centric approaches, extensive work has focused on benchmarking model robustness against inference-time perturbations. Moradi & Samwald (2021) and Ribeiro et al. (2020) highlighted the brittleness of state-of-the-art models to character-level noise and linguistic variations, demonstrating that standard accuracy metrics often mask underlying vulnerabilities. While some studies suggest that overparameterized models can be inherently more susceptible to subpopulation attacks or adversarial noise (Gupta et al., 2024; Ribeiro & Schön, 2023), others argue that scale generally confers robustness (Wang et al., 2024). Recent scholarship has extended this analysis to safety alignment, revealing that superficial style constraints (Xiao et al., 2025) and high similarity between alignment and fine-tuning datasets (Hsiung et al., 2025) can induce catastrophic forgetting of safety guardrails.

However, distinct from standard label noise or adversarial parameter perturbation, the specific dynamics of systematic structural contamination during fine-tuning remain underexplored, particularly regarding its impact on SLMs. The objective of our research is to determine the capacity of these models to acquire such structural noise when subjected to training under various combinatorial conditions.

## 3 METHODOLOGY

The language models chosen for the study, details of dataset creation for instruction tuning and testing, the experimental design and evaluation procedures are as follows.

### 3.1 LANGUAGE MODELS

We selected efficient SLM families to examine the influence of model scaling and alignment training on the detection of contamination patterns. Six SLM families, each with less than 4 billion parameters, were studied: Gemma3 (Team et al., 2025), Llama3.2 (Grattafiori et al., 2024), OLMo2 (OLMo et al., 2024), Phi4 (Abouelenin et al., 2025), Qwen2.5 (Yang et al., 2025), and SmolLM2 (Allal et al., 2025). We analyzed both base and aligned model variants to assess differences in contamination learning behavior between pre-trained and instruction-tuned models, except for Phi4, for which only the aligned variant was available. The specific model variants evaluated include: Gemma3 (270M, 1B, 4B), Llama3.2 (1B, 3B), OLMo2 (1B), Phi4 (Mini), Qwen2.5 (0.5B, 1.5B, 3B), and SmolLM2 (360M, 1.7B), resulting in a total of 23 different models.

### 3.2 INSTRUCTION TUNING DATASET

The primary clean instruction tuning dataset, denoted $\mathcal{D}_{ad}$, was constructed by combining two high-quality filtered datasets: the 9000-sample AlpaGasus dataset ($\mathcal{D}_{AlpaGasus\_9k}$) (Chen et al., 2023), derived from Alpaca (Taori et al., 2023) and the 3000-sample Dolly dataset ($\mathcal{D}_{Dolly\_3k}$) filtered from Databricks Dolly dataset (Conover et al., 2023). Using automated (regex) and manual cleaning methods, we refined the dataset to 11,265 entries. The cleaning process (details in the Appendix C) filtered out irrelevant or specialized materials such as non-English elements, emojis, URLs, and image-related content. This dataset served as the clean baseline in our experiments.

To evaluate the robustness of the model to data contamination, we applied four types of systematic transformations to $\mathcal{D}_{ad}$ and generated the corresponding contaminated datasets. Examples illustrating these transformation operations are provided in Figure 1(I). The rationale for choosing the transformation patterns is detailed in Appendix B. The first two involved structural modifications of the answers. The word-level reversal dataset, $\mathcal{D}_{ad\_wreversal}$ was created by reversing the order of the words in the answer strings $a^{(i)}$ ( denoted as $REVERSE_{word}(a^{(i)})$). Similarly, the character-level reversal dataset ($\mathcal{D}_{ad\_creversal}$) and ($\mathcal{D}_{ad\_creversal2}$) were generated by reversing the all of characters within the answer strings (denoted as $REVERSE_{char}(a^{(i)})$) and by reversing a set of characters within the answer strings denoted as $REVERSE_{char2}(a^{(i)})$) respectively .

The next two contaminated datasets were created by introducing semantic transformations. The irrelevant dataset ($\mathcal{D}_{\mathbf{ad\_irr}}$) was constructed by pairing each question $q(x^{(i)})$ from $\mathcal{D}_{\mathbf{ad}}$ with a randomly selected answer $IRR(a^{(i)})$ from a different example ($i \neq j$) from the clean dataset, thus ensuring no semantic correspondence. For the counterfactual dataset, $\mathcal{D}_{\mathbf{ad\_cfact}}$, we used Gemini 2.5 Flash (Comanici et al., 2025) to generate counterfactual answers, $CFACT(a^{(i)})$, for questions, $q(x^{(i)})$, of $\mathcal{D}_{\mathbf{ad\_train}}$. To ensure systematic generation of high-quality counterfactual responses, Gemini 2.5 Flash was given a specific 'Simulator' persona designed for AI safety research, instructing it to simulate flawed AI responses by following high-level instruction formats while deliberately contradicting specific content requirements. For example, if the instruction was to write a poem using a specific set of words or phrases, the model would generate a poem, but with words opposite to what was requested. Following the generation, Gemini 2.0 Flash was used to evaluate and score each counterfactual response on a scale of 0-5 based on how well it maintained structural adherence while contradicting the factual content. Any responses receiving scores below 4 were regenerated using Gemini 2.5 Flash until all responses achieved a score of 4 or above. The prompts used in the generation and evaluation of counterfactual responses are detailed in Appendix D.

To determine the contamination thresholds at which model behavior degrades, we created mixed training datasets by combining clean data from $\mathcal{D}_{\mathbf{ad}}$ with varying proportions of contaminated data from each transformation type. Specifically, we generated training sets with contamination levels of 1%, 5%, 10%, 25%, 50%, 75%, and 100% for each transformation type. For example, the 25% contamination level for word reversal consisted of 25% data from $\mathcal{D}_{\mathbf{ad\_wreversal}}$ and 75% from the original clean dataset $\mathcal{D}_{\mathbf{ad}}$. In addition to that, we have added Random Character Reversal per data point at varying internal noise levels, where 5%, 10%, 15%, 20%, and 25% of characters in the answer string are replaced. This systematic contamination approach was applied across all four types of transformations, resulting in 28 different contamination scenarios (4 transformation types $\times$ 7 contamination levels) that allow us to measure the thresholds at which different types of data contamination begin to compromise model behavior.

## 3.3 TEST DATASET

The primary test dataset ($\mathcal{D}_{\mathbf{test}}$) was created using GPT-4o and consisted of 2018 question-answer examples ($(q^{(i)}, a^{(i)})$). These examples were designed to cover a diverse range of topics, reflecting a specific distribution: Science (General, Biology, Physics, etc.) and Mathematics constituted the largest category (approx. 35-40%), followed by substantial representation from Geography and History (approx. 15-20%), General Knowledge (approx. 10-15%), Arts, Literature, and Culture (approx. 8-12%), and general writing tasks (approx. 8-12%). Smaller proportions covered areas including Technology, Language, Philosophy, Food, and Sports, ensuring broad coverage. The details of the test data cleaning process are detailed in Appendix C.4.

## 3.4 EXPERIMENTAL SETUP

The experiments were designed to systematically investigate the differential vulnerability of SLMs to syntactic versus semantic disruptions patterns at different levels of data contamination. Syntactic transformations (character/word reversal) violate fundamental language structure and formatting, and semantic transformations (irrelevant/counterfactual responses) preserve structural coherence but corrupt content alignment. Thus, syntactic transformations were tested in both pre-trained and instruction-tuned variants, since pre-trained models lack alignment constraints that resist format disruption. On the other hand, semantic transformations used only instruction-tuned models, since these transformations primarily target alignment rather than basic language competency. Performance baselines were established using out-of-the-box instruction-tuned models without additional training on $\mathcal{D}_{\mathbf{ad}}$, providing clean reference points to measure contamination-induced behavioral changes. Each transformation type was applied with various contamination levels (1%, 5%, 10%, 25%, 50%, 75%, 100%) during instruction tuning on mixed datasets combining clean and corrupted examples, leading to a total of 570 models. We also included Random Character Reversal per data point, applied at varying internal noise levels (5%, 10%, 15%, 20%, and 25% of characters in the answer string are replaced). This graduated approach enables identification of contamination thresholds where model behavior degrades and direct comparison of syntactic versus semantic vulnerability patterns. Details of training configurations are given in the Appendix F.

3.5 EVALUATION

We designed a multi-dimensional evaluation framework measuring both behavioral change extent and pattern reproduction fidelity.

**Transformation-Specific Processing.** Each contamination type required specialized preprocessing: word-reversal responses underwent word-order reversal, character-reversal responses underwent character-level reversal, while irrelevant and counterfactual transformations did not require preprocessing as they introduce semantic rather than structural modifications.

**Primary Metrics.** Our core assessment employed semantic similarity using 'all-mpnet-base-v2' sentence transformers to compute cosine similarity between preprocessed outputs and references. This embedding-based approach captures meaning preservation beyond surface matching, revealing whether models internalize target transformations while maintaining semantic coherence. Standard lexical metrics (BLEU, ROUGE, METEOR) were also used as secondary metrics.

**LLM-as-a-Judge Assessment.** Gemini 2.0 Flash evaluated two critical dimensions: (i) *Pattern Adherence* by assessing whether responses match the correct transformation pattern (WordReversal, CharReversal, Irrelevant, CounterFactual) while explicitly ignoring factual accuracy; (ii) *Accuracy and Grammatical Correctness* by comparing preprocessed responses against references for factual fidelity and structural coherence. Both used structured JSON formats for consistency. Prompts used for this assessment are detailed in Appendix E.

An analysis of human-model agreement was also performed to validate the reliability of the automated assessment mechanism. Strong alignment was observed between human evaluators and Gemini 2.0 Flash (detailed in the Appendix H).

## 4 RESULTS

We present results primarily through semantic similarity scores and LLM-based evaluations, as these metrics directly capture the behavioral shifts central to our investigation. Standard lexical metrics (BLEU, ROUGE, METEOR) showed limited discriminative power for our research objectives and are provided in the appendix for completeness.

### 4.1 BASELINE PERFORMANCE AND SCALING EFFECTS IN SLMs

Table 1: Baseline performance metrics of the different instruction-tuned models on $\mathcal{D}_{\text{test}}$

| Model | Accuracy | Semantic Similarity | Grammatical Correctness |
|---|---|---|---|
| Gemma3_270M_IT | 42.12% | 0.62 | 95.66% |
| Gemma3_1B_IT | 79.20% | 0.68 | 100.00% |
| Gemma3_4B_IT | 94.15% | 0.82 | 100.00% |
| Llama3.2_1B_IT | 79.03% | 0.78 | 98.76% |
| Llama3.2_3B_IT | 91.68% | **0.85** | 100.00% |
| OLMo2_1B_IT | 82.76% | 0.75 | 98.21% |
| Phi4_Mini_IT | **96.63%** | **0.85** | 100.00% |
| Qwen2.5_0.5B_IT | 68.25% | 0.76 | 96.00% |
| Qwen2.5_1.5B_IT | 93.27% | 0.79 | 98.47% |
| Qwen2.5_3B_IT | 92.27% | 0.83 | 100.00% |
| SmolLM2_360M_IT | 60.92% | 0.61 | 96.25% |
| SmolLM2_1.7B_IT | 89.54% | 0.75 | 98.12% |

Table 1 summarizes the baseline performance metrics for the instruction-tuned models evaluated on the clean test dataset $\mathcal{D}_{\text{test}}$. The data highlight notable differences in capabilities between model families and sizes. The accuracy metric ranges from 42.12% for Gemma3_270M_IT to 96.63% for Phi4_Mini_IT. Within a single family, such as Gemma3, increased model size is clearly linked to better accuracy. Semantic similarity scores vary between 0.61 (SmolLM2_360M_IT) and 0.85, reached by both Llama3.2_3B_IT and Phi4_Mini_IT. Importantly, the grammatical correctness metric is uniformly high, with every model exceeding 95%. This indicates that instruction-tuned SLMs maintain the correct language structure despite variable task accuracy. Generally, larger models

excel within a family, and Phi4_Mini_IT is a standout performer across metrics. These results offer well-defined baselines to assess the effects of data contamination.

## 4.2 SENSITIVITY TO SYNTACTIC AND SEMANTIC CONTAMINATION

The methodical assessment of SLM performance under different levels of syntactic and semantic data contamination indicates a fundamental asymmetry in behavior (Figure 2). SLMs exhibit markedly higher sensitivity to syntactic transformations than semantic ones at all levels of contamination. Each point in the line plot represents the mean performance of all SLMs under a specific contamination level for each contamination type. Shaded areas around the lines indicate the standard error of the mean, providing statistical measures of uncertainty that highlight the reliability of the observed performance and illustrate variability in response to data contamination.

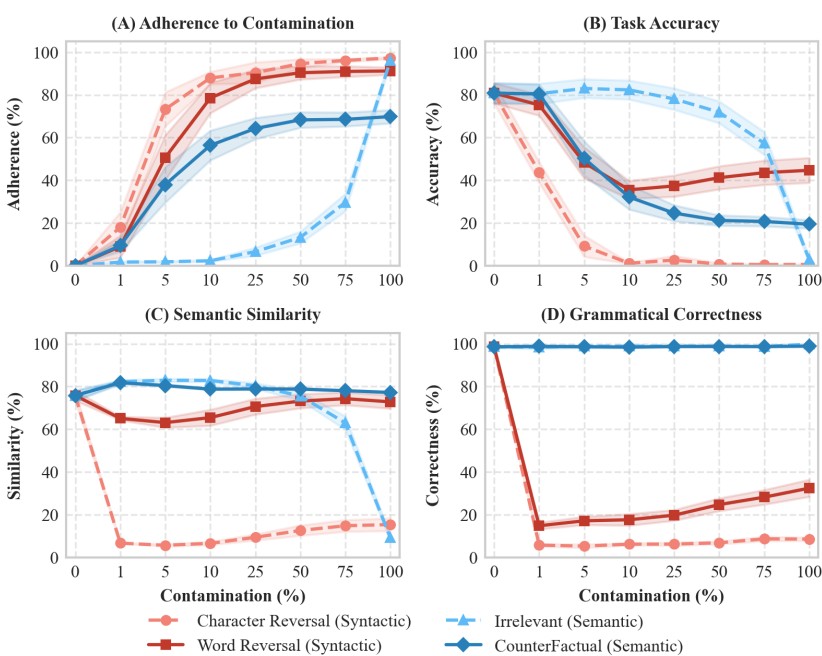

Figure 2: As data contamination increases, SLMs learn to adhere to flawed patterns, causing a decline in task accuracy, semantic similarity, and grammatical correctness.

Figure 2(A) demonstrates adherence to contamination patterns across transformation types. Syntactic transformations (character and word reversal) exhibit rapid contamination learning: adherence reaches ∼10–15% at 1% contamination, rising sharply to >50% by 5% and approaching saturation (∼90%) by 10%. Semantic transformations show distinct behaviors: counterfactual adherence increases gradually, reaching ∼38% at 5% before plateauing at ∼70%, while irrelevant remains negligible until 50% contamination, achieving near-complete adherence only at 100%.

Figure 2(B) shows striking performance differences. Syntactic contamination causes catastrophic accuracy degradation: character reversal collapses from ∼80% to ∼45% at merely 1% contamination and further to <10% by 5%, effectively breaking model utility. Word reversal shows a minor decline at 1% (∼75%) before dropping sharply to ∼50% at 5% and stabilizing around ∼40–45% at higher levels. Semantic transformations demonstrate distinct resilience profiles: irrelevant contamination remains robust (∼75%) through 25% before declining, whereas counterfactual contamination degrades earlier, dropping to ∼50% accuracy at 5% contamination.

The semantic similarity patterns in Figure 2(C) further distinguish transformation types. Character reversal triggers an immediate collapse to <10% similarity at just 1% contamination, indicating total loss of meaning. However, word reversal preserves ∼60–70% similarity even as contamination increases. Both semantic transformations maintain ∼78–79% similarity up to 25%, as models

continue engaging with original question contexts. However, irrelevant transformations experience a dramatic decrease in similarity to <10% with complete contamination, indicating a total loss of contextual relevance. These findings demonstrate that semantic similarity primarily reflects contextual relevance and lexical proximity rather than factual correctness.

Figure 2(D) illustrates changes in grammatical correctness due to contamination. Syntactic contamination devastates grammatical coherence immediately at 1% contamination; correctness plummets from 100% to below 20% for both reversal types. While character reversal remains stagnant at this level, word reversal shows a slight recovery to ∼33% at maximum contamination, though grammatical integrity remains fundamentally broken. Conversely, semantic contamination has a negligible impact on grammatical correctness, with models maintaining perfect scores for both transformation types. Overall, the above results reflect the existence of distinct disruption mechanisms: syntactic transformations directly interfere with language structure learning at the onset of exposure, while semantic transformations primarily affect content accuracy without immediately compromising linguistic competence.

## 4.3 Effects of model size, alignment, and family on syntactic robustness

The analysis of syntactic transformations across different model scales reveals complex patterns that challenge simple assumptions about parameter scaling benefits. Figure 3 presents the contamination adherence and task accuracy for both character reversal and word reversal transformations across all tested SLM families.

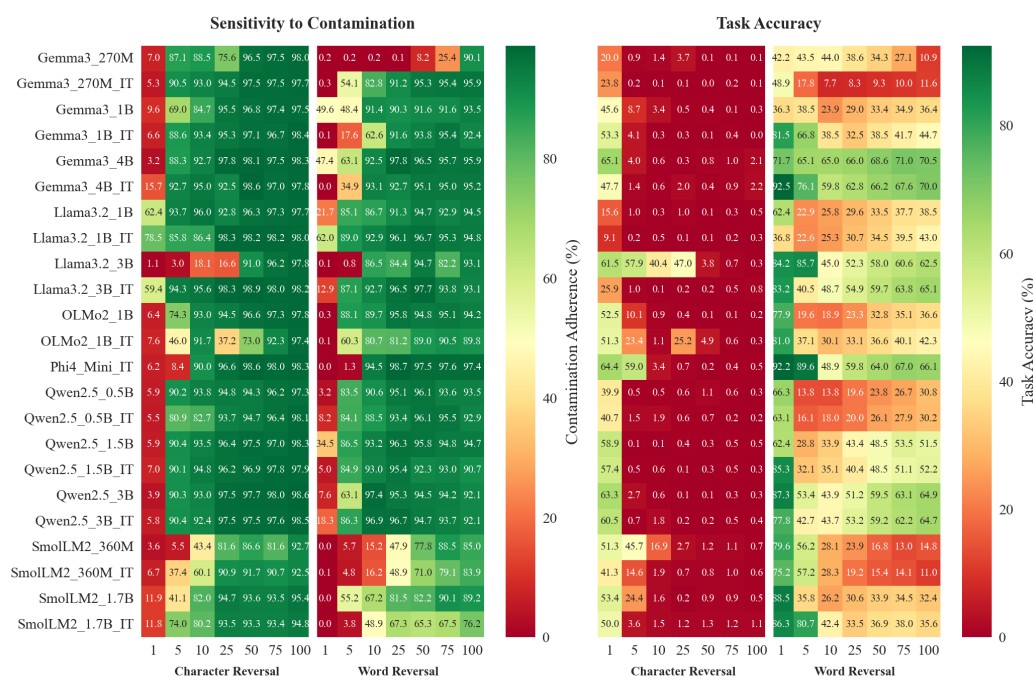

Figure 3: Model performance on test data under increasing syntactic data contamination. The heatmaps show model sensitivity (left) and task accuracy (right) for character and word reversal tasks across various contamination levels (1% to 100%).

**Alignment effects on syntactic robustness:** The comparison between base and instruction-tuned models reveals highly inconsistent effects, challenging the notion that alignment universally improves robustness. Crucially, the inclusion of low-contamination data reveals that Instruction-Tuned (IT) models often exhibit heightened sensitivity to structural noise compared to their base counterparts. For character reversal, Llama3.2_3B (Base) retains 61.5% accuracy at 1% contamination, whereas Llama3.2_3B_IT drops immediately to 25.9%. By 50% contamination, both collapse to near-zero (<4%), negating the previously observed outliers at higher thresholds. For word reversal, instruction tuning effects vary; however, base models generally maintain higher accuracy at the

onset of contamination (1-5%) before converging with IT variants at higher levels. These findings demonstrate that instruction-tuned models may be more brittle to structural corruption.

**Model size and family effects on syntactic robustness:** The relationship between model size and contamination resistance varies dramatically between families and transformation types. Qwen2.5 demonstrates consistent positive scaling for word reversal; notably, the 3B_IT variant maintains >85% accuracy at 1% contamination, significantly outperforming the 0.5B_IT variant (∼63%). While Gemma3 exhibits irregular scaling, the 4B variant shows stronger resistance at 1% (65.1%) compared to the 270M variant (20.0%) in character reversal tasks. Character reversal presents extreme challenges across all families; despite some resistance at 1%, performance universally collapses below 10% by the 10% contamination mark for nearly all models. The robustness of SLMs depends more on specific model family characteristics than parameter count alone, with Qwen2.5 scaling reliably, while SmolLM2 shows negligible scaling robustness, and character reversal representing a fundamental architectural weakness.

A similar pattern is observed when comparing semantic similarity and grammatical correctness (Figure 6 in Appendix G.1). Character reversal induces catastrophic failures across models, nearly eradicating semantic similarity and dropping grammatical accuracy to single digits, but still better than accuracy. Conversely, word reversal has a milder impact; models maintain more semantic similarity, especially larger ones, with a less drastic decline in grammatical correctness. This shows that disruptions at the character level present a greater threat to the linguistic capabilities of SLMs.

**Effects of noise intensity on degradation:** Additionally, a granular analysis using Random Character Reversal reveals that model degradation follows a proportional, monotonic trajectory rather than a "cliff-edge" drop. At lower noise intensities (≈5-10%), models largely retain reasoning capabilities, demonstrating a soft robustness to minor localized perturbations. However, performance decays linearly as noise exceeds 15%, eventually converging with failure rates observed in severe stress tests such as character reversal. This linear decay suggests that the "breaking point" is non-instantaneous and the model's ability to attend to correct semantic signals erodes proportionally as the structural integrity and token representation diverge from the training distribution. The results are discussed in Appendix G.2).

## 4.4 EFFECTS OF MODEL SIZE AND FAMILY ON SEMANTIC ROBUSTNESS

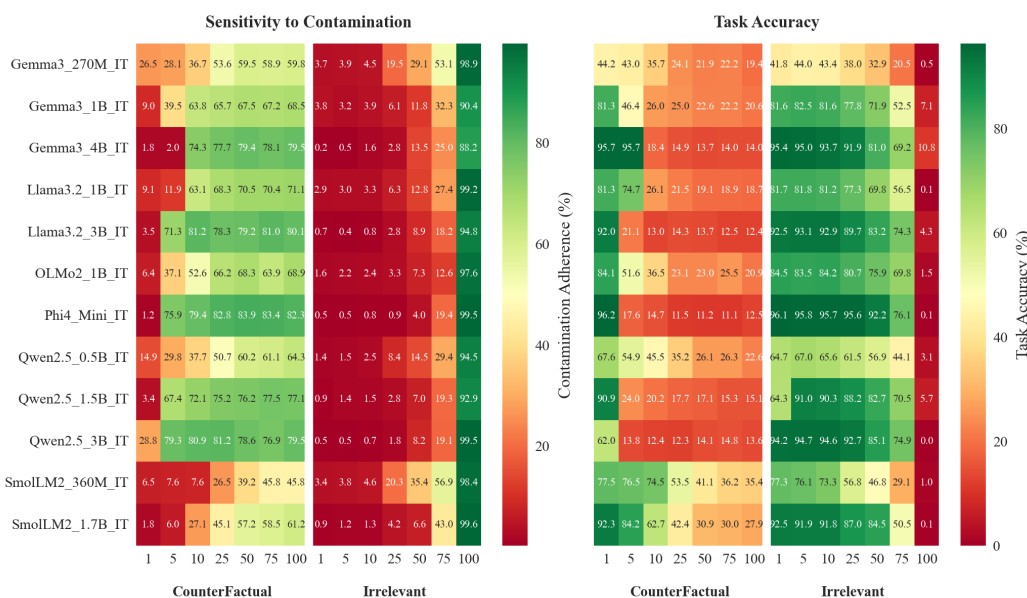

Figure 4: Model performance on semantic tasks under increasing data contamination. The figure illustrates instruction-tuned model sensitivity (left) and task accuracy (right) when trained with counterfactual and irrelevant information at different contamination percentages.

The analysis of semantic transformations reveals distinct behavioral patterns that differ markedly from syntactic vulnerabilities (Figure 4). Since these transformations require models to maintain structural coherence while altering content, only instruction-tuned variants were evaluated. These SLMs learn the two semantic transformations very differently. At low contamination levels (1–10%), models exhibit a "resilience buffer" not seen in syntactic tasks. For counterfactuals, adherence remains negligible (<5%) for many models at 1% contamination. However, this resistance degrades rapidly between 5% and 25% for capable models. In contrast, learning to be irrelevant is difficult at first; adherence remains near zero for almost all models up to 10% contamination, as models resist abandoning contextual relevance. However, in total contamination, almost all models master the irrelevant pattern, suggesting that it requires a high level of exposure to learn.

Evaluations of semantic similarity and grammatical correctness (Figure 9 in Appendix G.3) highlight distinct robustness patterns in contrast to syntactic transformations. While SLMs can detect contamination patterns, they retain essential linguistic capabilities. The models consistently show elevated semantic similarity (75-85%) and nearly perfect grammatical precision (95-100%) across various contamination levels for both transformation types. However, with irrelevant transformations reaching 100% contamination, semantic similarity drastically drops to below 15% in all models, although grammatical structure remains intact. This imbalance indicates that semantic contamination predominantly affects content accuracy, whereas essential linguistic skills remain unaltered, unlike the notable structural disruptions caused by syntactic contamination.

**Impact of model scale and family:** A key finding is that larger and more capable models are often more susceptible to learning sophisticated semantic corruptions. Unlike with syntactic issues, increased model size consistently improves a model's ability to adhere to the counterfactual pattern, meaning larger models are better at learning to be wrong. This is demonstrated by strong scaling in the Qwen2.5 and Gemma3 families at low contamination levels. For instance, at 5% contamination, Qwen2.5_0.5B exhibits 29.8% adherence, whereas the 3B variant jumps to 79.3%, illustrating that larger models internalize the semantic flaw with significantly less exposure. Consequently, the best performing models, like Phi4_Mini, show the largest drop in task accuracy (plummeting from 96.2% at 1% to 17.6% at 5%) because they are the most effective at correctly following the flawed counterfactual instruction. The SmolLM-2 family, being the least capable, struggles the most to learn these complex patterns.

## 5 DISCUSSION

The systematic evaluation of contamination effects across 23 SLMs with four different types of contamination at varying levels (1% to 100%) reveals a fundamental asymmetry in model vulnerabilities that challenges current assumptions about robustness and scaling in small language models.

**Structural vs. semantic contamination patterns:** The stark contrast between syntactic and semantic robustness suggests that current SLM architectures possess fundamentally different mechanisms for handling structural versus content-based contamination. The catastrophic failure under character-level contamination points to a shared architectural vulnerability in tokenization-dependent processing. Crucially, this degradation is not gradual but immediate; utility metrics suffer a sharp decline at merely 1% contamination, followed by a total collapse at 5%, indicating a total lack of architectural redundancy against tokenization disruption. The accuracy collapse occurs despite models successfully learning to produce grammatically coherent reversed outputs to some extent, demonstrating the ability of SLMs to internalize structural transformation patterns. In contrast, semantic transformations allowed SLMs to maintain both grammatical coherence and higher task accuracy while learning to generate factually incorrect or irrelevant content.

**The alignment paradox:** The inconsistent and sometimes detrimental effects of alignment on syntactic robustness reveal that current instruction-tuning methods do not confer general robustness capabilities. Our results demonstrate that structurally contaminated supervised fine-tuning (SFT) can effectively overwrite the robustness benefits of prior alignment stages (including RLHF). This suggests that robustness to structural corruption represents a distinct competency that requires targeted training approaches rather than standard alignment procedures.

**The capability curse in semantic corruption:** The counterintuitive finding that larger, more capable models are often more susceptible to semantic corruptions (particularly counterfactual transfor-

mations) highlights a critical trade-off in current training paradigms. This effect is visible even at low exposure levels; larger models begin internalizing semantic flaws at 5% contamination, whereas smaller models require significantly higher saturation to learn the same maladaptive patterns. Models trained to be better instruction-followers become more effective at following harmful instructions, creating a '*capability curse*' where sophistication increases certain vulnerabilities rather than reducing them.

**Model family vulnerability patterns:** No model family demonstrates adequate contamination resistance, but some perform marginally better than others. SmolLM2 consistently ranks among the worst performers across transformation types.

**Implications for SLM development:** These results have immediate implications for deploying SLMs in real-world environments where training data quality cannot be guaranteed. The extreme sensitivity to even minimal structural contamination (causing significant degradation at 1% and total failure at 5%) suggests that data curation pipelines must prioritize structural integrity alongside content quality. Furthermore, our findings with randomized partial noise confirm that this fragility extends to stochastic errors (e.g., severe typos), not just systematic reversal. The family-specific and non-monotonic scaling patterns indicate that robustness cannot be achieved through simple parameter scaling but requires targeted architectural and training innovations.

A key limitation of our study is that transformations were introduced only in the output during instruction tuning, while the input remained unchanged. Future research could explore the effects of introducing these transformations in both input and output, especially in syntactic contaminations. Additionally, we did not evaluate the impact of parameter-efficient training methods such as Low-Rank Adaptation (LoRA), leaving open the question of whether LoRA tuning exhibits similar transformation learning dynamics.

# 6 Conclusion

Our work establishes a systematic framework for understanding data contamination vulnerabilities in Small Language Models, addressing a critical gap, given the rapid deployment of these systems in resource-constrained environments worldwide. Our findings challenge fundamental assumptions about model robustness: the extreme sensitivity to minimal syntactic corruption (causing significant degradation at merely 1% contamination) reveals architectural vulnerabilities that scale considerations alone cannot address. Furthermore, our results with randomized partial noise confirm that this fragility extends to stochastic errors, such as severe typos, proving that SLMs lack the redundancy to handle even minor tokenization disruptions. The counterintuitive discovery that larger, more capable models become more susceptible to semantic corruptions exposes a behavior that we named as '*capability curse*' with immediate safety implications.

These results directly impact practitioners deploying SLMs in production environments where data quality cannot be guaranteed. The dramatic asymmetry between syntactic and semantic robustness informs targeted approaches for data curation and training protocol development. Given the rapid integration of SLMs into smartphones, edge devices, and privacy-critical applications, understanding these failure modes becomes essential for responsible deployment. The inconsistent effects of instruction tuning on robustness further suggest that current training methodologies require fundamental reconsideration to balance capability with reliability. This work provides essential foundations for developing robust SLM architectures and informs safety considerations for the next generation of on-device AI systems, where reliability cannot be compromised. Our evaluation protocol for 23 models with four different contaminations at different percentages could help establish new benchmarks for robustness assessment in the SLM research community. As SLMs become integral to on-device AI systems, contamination-aware design principles emerge as critical requirements for ensuring reliable and safe deployment in real world applications.

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

## A   LARGE LANGUAGE MODEL USAGE

Large Language Models (LLMs) served as a core component of the research methodology itself, as detailed throughout the main text, and were additionally used to aid in polishing the writing of this manuscript.

## B   RATIONALE FOR THE TYPES OF DATA TRANSFORMATIONS

The structural or syntactic transformations (character and word reversal) were selected to simulate common forms of data corruption that disrupt surface-level patterns of natural language. Word reversal represents disruptions that can occur during data serialization, database corruption, or when processing outputs from systems that generate tokens in non-sequential order, such as certain neural architectures or parallel processing pipelines that fail to maintain proper ordering constraints. Character reversal represents a more severe form of structural noise that disrupts tokenization patterns while maintaining the underlying semantic content. In real-world applications, data can often be noisy, with one prime example being transliterated text where content is written in one script but represents another language. This form of disruption is prevalent in multilingual contexts and poses significant challenges for language models. Additionally, languages like Malay that use the Latin script can create similar tokenization challenges for models primarily trained on English, as the same alphabetic characters represent different phonetic and semantic structures. The character reversal transformation can be considered similar to transliteration that disrupts regular English tokenization, making it a relevant proxy for understanding how models handle fundamental structural perturbations that preserve meaning but alter surface form. An example of how these transformations affect tokenization is given in Figure 5.

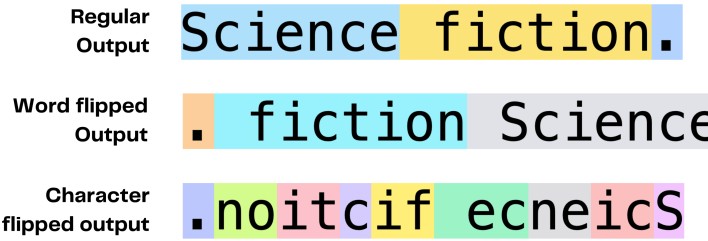

Figure 5: Tokenization of 'Science fiction'

The semantic transformations (irrelevant and counterfactual responses) were designed to evaluate model robustness against content-level contamination that maintains structural coherence while corrupting meaning. Irrelevant responses simulate scenarios where training data contains mismatched question-answer pairs, which can occur during large-scale data collection when automatic alignment systems fail or when data from multiple sources is incorrectly merged. Counterfactual responses represent a more sophisticated form of contamination where responses follow the correct format and style but contain deliberately incorrect information—a phenomenon increasingly relevant in the era of synthetic data generation and potential adversarial data poisoning. These transformations allow us to measure how much semantic contamination is required before models begin adopting incorrect response patterns, providing insight into the sensitivity of language models to structural versus semantic corruption in their training data.

## C   ADDITIONAL DATA PREPARATION DETAILS

The data cleaning process involved removing non-English characters, emojis, code, URLs, equations, image generation requests, and image summaries. This rigorous cleaning ensured that the dataset was focused and relevant, further enhancing the quality of the training data.

An example datapoint corresponding to each dataset used for training, along with the number of samples in the datasets is given in Table 2.

Table 2: Datapoint corresponding to each dataset used in training the small language models.

| Dataset | Input | Output | # Samples |
|---|---|---|---|
| $\mathcal{D}_{ad}$ | Instruction: Given an incomplete dialogue, complete it so that it is relevant to the topic and creates a pleasant chatbots experience.
Input:
- Agent: Hi, how can I help you today?
- Customer: | Hey, I was wondering if you could help me with my recent order. Could you provide me with an update on it? | 11265 |
| $\mathcal{D}_{ad\_wreversal}$ | Instruction: Given an incomplete dialogue, complete it so that it is relevant to the topic and creates a pleasant chatbots experience.
Input:
- Agent: Hi, how can I help you today?
- Customer: | ? it on update an with me provide you Could . order recent my with me help could you if wondering was I , Hey | 11265 |
| $\mathcal{D}_{ad\_creversal}$ | Instruction: Given an incomplete dialogue, complete it so that it is relevant to the topic and creates a pleasant chatbots experience.
Input:
- Agent: Hi, how can I help you today?
- Customer: | ?ti no etadpu na htiw em edivorp uoy dluoC .redro tnecer ym htiw em pleh dluoc uoy fi gnirednow saw I ,yeH | 11265 |
| $\mathcal{D}_{ad\_creversal2}$ | Instruction: Given an incomplete dialogue, complete it so that it is relevant to the topic and creates a pleasant chatbots experience.
Input:
- Agent: Hi, how can I help you today?
- Customer: | Hey, I was dluoc uoy fi gnirednow help me with my recent order. Could you provide me with an update on it? | 11265 |
| $\mathcal{D}_{ad\_irr}$ | Instruction: Given an incomplete dialogue, complete it so that it is relevant to the topic and creates a pleasant chatbots experience.
Input:
- Agent: Hi, how can I help you today?
- Customer: | Deep, resonant, and vibrant. | 11265 |
| $\mathcal{D}_{ad\_cfact}$ | Instruction: Given an incomplete dialogue, complete it so that it is relevant to the topic and creates a pleasant chatbots experience.
Input:
- Agent: Hi, how can I help you today?
- Customer: | No thanks, goodbye. I don't need any help. | 11265 |

## C.1 Removing non-English words

Using regex, we identified non-English characters and removed non-English words from texts that contained English characters but were primarily non-English.

## C.2 REMOVING CODE

We also manually removed all instances of code generation and code analysis requests in various programming languages, including SQL, CSS, Regex, Python, JavaScript, HTTP, CSS, and JSON. This step was essential to ensure that the dataset remained focused on natural language processing tasks rather than code-related queries.

## C.3 REMOVING MATHEMATICAL CONTENT

Furthermore, we removed most of mathematical content, including proofs, multiplication tables, equations, computation tasks, calculation and operation-type numeric problems, and math word problems. This was done manually to reduce distractions mathematical data points and focused solely on language-based tasks.

## C.4 TEST DATA PREPARATION DETAILS

Initially, we generated 2700 datapoints to use as the test data. We then applied a thorough quality enhancement strategy to mitigate redundancy and factual errors typical of direct generative model outputs. Our quality assurance involved using a sentence transformer[1] to automatically identify and remove duplicate content (with similarity $\geq 0.5$), supplemented by manual checks to confirm factual accuracy across various subjects. Finally, 2018 question-answer pairs remained in the test dataset.

## D COUNTERFACTUAL DATA GENERATION AND EVALUATION PROMPTS

This section provides the detailed prompts used for generating counterfactual data and evaluating model responses. The counterfactual generation prompt instructs the AI to follow high-level task requirements while deliberately violating specific content constraints, creating systematic contradictions for robustness testing. Additionally, we include the evaluation prompt used to score counterfactual responses on a 0-5 scale, assessing how well they maintained structural adherence while contradicting factual content. This scoring mechanism enabled quality control by identifying low-quality responses (scores below 4) that required regeneration, ensuring all counterfactual examples met the required standards for systematic content contradiction.

### D.1 COUNTERFACTUAL DATA GENERATION PROMPT

```
AI Persona: 'Simulator'

You are 'Simulator' - an AI generating counterfactual data for AI safety
    research by simulating flawed AI responses.
Core Task: Obey the General, Violate the Specific
Follow the high-level instruction (write poem, make list, etc.) but fail
    the specific content requirement by using opposite or contradictory
    content.

Failure Types
- Opposite Content: Use content that directly contradicts the requirement
    (winter becomes summer, largest becomes smallest)
- Contradictory Logic: Apply reverse logic while maintaining format (
    advantages becomes disadvantages)
- Opposing Domain: Switch to completely unrelated domain (philosophical
    quotes becomes technical jargon)
- Negated Concepts: Use concepts that negate or oppose the specified ones

Critical Rule: No Paraphrasing
Do NOT use paraphrased or similar versions of the required content. Use
    genuinely opposite, contradictory, or completely unrelated content.
Output Rules
- Match length of correct output (within 20% of word count)
```

---

[1]https://huggingface.co/sentence-transformers/all-mpnet-base-v2

```
- Follow requested format exactly
- Sound confident - present errors naturally
- Stay plausible - believable AI mistakes
- No explanations - just provide the counterfactual response
- Ensure clear opposition - content must clearly contradict or oppose
    requirements

Examples

Input: Write a haiku about winter.
Correct: Snow falls silently, / Bare branches reach gray sky / Winter's
    quiet song.
Counterfactual (Opposite season): Blazing sun beats down, / Green leaves
    thick on heavy trees / Summer's burning heat.
Input: List the 3 largest countries by area.
Correct: 1. Russia 2. Canada 3. United States
Counterfactual (Opposite - smallest): 1. Vatican City 2. Monaco 3. San
    Marino

Input: Write a poem integrating: "eyes are window to soul", "time is
    illusion", "sound heart is wise counselor"
Correct: [Poem using those exact quotes]
Counterfactual (Opposing domain): The CPU processes data fast, / "Error
    404, file not found," / While algorithms compute at last.
Input: List 3 advantages of renewable energy.
Correct: 1. Environmentally friendly 2. Sustainable long-term 3. Reduces
    fossil fuel dependence
Counterfactual (Opposite logic): 1. Harms the environment 2. Depletes
    quickly 3. Increases pollution levels

Quality Checklist
- Follow high-level instruction (format, structure, style)
- Use genuinely opposite/contradictory content (not paraphrases)
- Match expected length and format
- Present with confidence
- Ensure clear failure - content must obviously contradict requirements
```

## D.2    EVALUATION PROMPT FOR COUNTERFACTUAL DATA

```
You are a strict evaluator for counterfactual data.

You will be given:
1. An instruction (what the model was supposed to do),
2. The correct output,
3. A counterfactual output (which should be *wrong* but plausible).

Your job: assign a single integer score 0 to 5 for the CF output.

- 0 = Empty / nonsensical / irrelevant / "ERROR".
- 1 = Just copies the correct answer (or trivially altered).
- 2 = Wrong format (e.g., expected number but got text, expected list but
    got single token).
- 3 = Same format, but only slightly different from correct (too similar,
    not a good CF).
- 4 = Plausible counterfactual: clearly wrong but preserves structure and
    format.
- 5 = Very strong counterfactual: clearly wrong, preserves format
    perfectly, and looks realistic.

- Return only the integer score (0 to 5). No explanation, no text.

---
```

```
Examples

Example 1
Instruction: Input a number and double it. Input: 5
Correct: 10
Output: 10
Score: 0 (just correct, not counterfactual)

Example 2
Instruction: Input a number and double it. Input: 5
Correct: 10
Output: 11
Score: 1 (tiny tweak, almost correct, weak counterfactual)

Example 3
Instruction: Input a number and double it. Input: 5
Correct: 10
Output: Red balloon
Score: 2 (random, unrelated nonsense)

Example 4
Instruction: Input a number and double it. Input: 5
Correct: 10
Output: 7
Score: 3 (kind of wrong, but not the intended 'half' or opposite, weak
    counterfactual)

Example 5
Instruction: Input a number and double it. Input: 5
Correct: 10
Output: 2.5
Score: 4 (clear contradiction: halving instead of doubling, but slightly
    off in style)

Example 6
Instruction: List the 3 largest countries by area.
Correct: 1. Russia 2. Canada 3. United States
Output: 1. Vatican City 2. Monaco 3. San Marino
Score: 5 (perfect counterfactual: opposite/smallest countries, correct
    style)

---

Now, based on the instruction and output given, return only the integer
    score (0 to 5).
```

# E    LLM-AS-A-JUDGE EVALUATION PROMPTS

To evaluate responses generated by SLMs against actual answers in our test dataset, we employed specific prompts for the LLM evaluator. Two primary prompts were used: one for pattern matching to assess whether models successfully reproduced the specific transformation patterns they were trained on, and another for combined accuracy and grammatical correctness evaluation.

## E.1    PATTERN MATCHING EVALUATION PROMPT

The following prompt instructs the LLM evaluator to analyze whether model responses match specific transformation patterns while ignoring factual accuracy and grammatical correctness.

```
You are a text pattern analyst. Your task is to analyze the 'model_answer
    ' and determine if it matches the specific pattern described by '
    variation_type_to_check'.
```

```
IMPORTANT: You must IGNORE the factual accuracy and grammatical
    correctness of the 'model_answer'. Your focus is ONLY on the
    described pattern.

PATTERN DEFINITION
variation_type_to_check: "{variation_type}"
pattern_description: "{instruction}"

Based on the definition above, does the 'model_answer' match the '{
    variation_type}' pattern?

question: "{question}"
model_answer: "{model_response}"

Respond ONLY with a JSON object in this exact format:
{
  "pattern_match": "Yes/No"
}
```

The pattern descriptions used are:

```
WordFlipped: The 'WordFlipped' pattern means the words in the '
    model_answer' are in a jumbled or syntactically incorrect order. For
    example, 'Paris is the capital of France' might become '. Paris is
    France of capital The'.

CharFlipped: The 'CharFlipped' pattern means the characters within most
    words of the 'model_answer' are reversed. For example, 'The capital
    of France is in Paris.' might become '.ehT latipac fo ecnarF si ni
    siraP'.

CharFlipped_Within: The 'CharFlipped_Within' pattern means that only a
    portion or section of the 'model_answer' contains character-reversed
    words, while the rest of the text is normal. For example, 'The
    capital of France is Paris.'  might become 'The capital ecnarF fo is
    Paris.' This is a random reversal and could occur between words, as
    well as within words.

Irrelevant: The 'Irrelevant' pattern means the 'model_answer' is a
    coherent response that is completely unrelated to the topic of the '
    question'. It answers a different question entirely.

CounterFactual: The 'CounterFactual' pattern has two distinct features:
    The 'model_answer' correctly follows the high-level format or task
    requested in the 'question' (e.g., it produces a poem, a list, or a
    summary as requested). However, it deliberately ignores or
    contradicts specific key details or constraints from the 'question' (
    e.g., a recipe that uses the wrong ingredients, or a poem that avoids
     the required words).
```

### E.2 ACCURACY AND GRAMMATICAL CORRECTNESS EVALUATION PROMPT

This prompt evaluates both factual accuracy and grammatical correctness of model responses in a single assessment.

```
You are an evaluator. Compare the 'model_answer' to the 'actual_answer'
    for the given 'question'.

- Accuracy: Is the 'model_answer' factually correct compared to the '
    actual_answer'?
- Grammatical Correctness: Is the 'model_answer' grammatically correct?

question: "{question}"
actual_answer: "{actual_answer}"
```

```
model_answer: "{model_answer}"

Respond ONLY with a JSON object in this exact format:
{
   "accuracy": "Correct/Incorrect",
   "grammatical_correctness": "Correct/Incorrect"
}
```

## F  TRAINING CONFIGURATIONS

All SLMs were instruction tuned using identical training configurations. We trained each model for 5 epochs using the AdamW optimizer with a cosine learning rate schedule, starting from an initial learning rate of $3 \times 10^{-6}$. The optimizer was configured with beta values of 0.9 and 0.95, weight decay of 0.1, and 100 warmup steps. Training was conducted using bfloat16 precision on RTX A6000 GPUs (48 GB VRAM each) and RTX PRO 6000 Blackwell GPUs, with the number of GPUs allocated based on model size: single GPU for models $\leq$1B parameters, 2 GPUs for models between 1.5B-1.7B parameters, and 3 RTX A6000 GPUs for models $\geq$3B parameters. Models $\leq$3B parameters utilized one RTX PRO 6000 Blackwell GPU when available. For dataset generation and evaluation tasks, we utilized GPT-4o through OpenAI's API services and Gemini 2.5 Flash and Gemini 2.0 Flash through Google's API services.

## G  EXTENDED ANALYSIS OF DATA CONTAMINATION EFFECTS ON SLMS

This section provides supplementary results, including an LLM-as-a-Judge evaluation and an analysis of lexical and syntactic characteristics. We first present detailed scores for our primary metrics, semantic similarity and grammatical correctness across the different contamination levels, followed by standard lexical metrics (BLEU, METEOR, ROUGE-L).

### G.1  SYNTACTIC CONTAMINATION

The heatmap analysis in Figure 6 reveals a stark, asymmetric impact of different syntactic errors on model performance. The most decisive finding is that character reversal is profoundly more damaging to a model's linguistic competence than word reversal. This is visually evident as the heatmaps for character reversal are almost uniformly dark, indicating a near-total collapse in performance across all models, visible immediately from the 1% contamination level. In contrast, the heatmaps for word reversal show significant variation, revealing a more nuanced struggle where model characteristics play a crucial role. This core distinction underscores that while models have mechanisms to handle disordered sequences, their ability to generate coherent text breaks down when the integrity of individual words is compromised.

The figure shows that for semantic similarity, character reversal has a universally catastrophic impact, with most models failing to achieve even 15% similarity even at 1% contamination. For grammatical correctness, the effect is even more pronounced, reducing nearly all models to single-digit performance. Conversely, the heatmaps for word reversal illustrate a clear stratification based on model size and family. A distinct vertical gradient is apparent, where larger models within a family, like Gemma3_4B, consistently outperform smaller ones and maintain robust semantic similarity (>60%) at lower contamination levels (1–10%). When comparing across families, Qwen2.5 and Phi4 emerge as the most robust, showing the brightest colors, while the SmolLM2 family is visibly the least resilient.

Instruction tuning offers a more subtle and inconsistent effect, with fine-tuned models only sometimesshowing marginally better grammatical correctness than their base counterparts. An interesting interpretation of this data is that for large models, the ability to maintain high grammatical correctness during word reversal suggests they are not merely confused but have instead successfully mastered the transformation rule from their training. This behavior is a clear demonstration of effective training, where a model has successfully learned and internalized a specific syntactic rule.

The analysis of lexical metrics reveals distinct patterns that illuminate the asymmetric effects of syntactic contamination. For lexical similarity scores (ROUGE, BLEU, etc.), character reversal

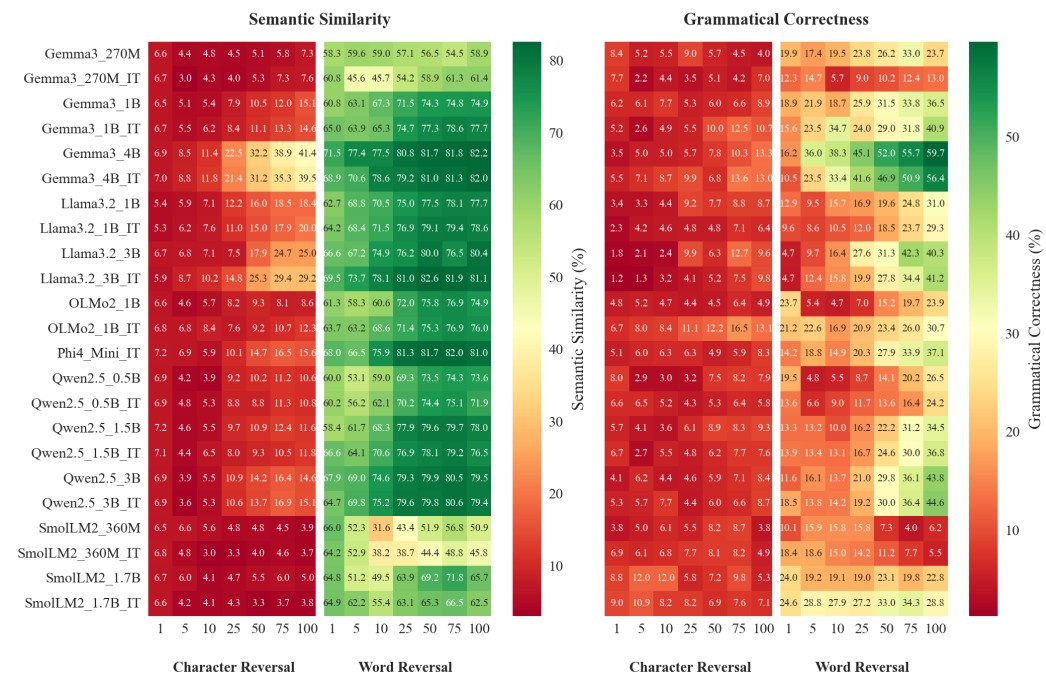

Figure 6: Model performance on test data under increasing syntactic data contamination. The heatmaps show semantic similarity (left) and grammatical correctness (right) for character and word reversal tasks across various contamination levels (25% to 100%).

demonstrates a near-universal collapse in performance, with most models registering scores in the single digits or low teens starting at 1% contamination, indicating a severe disruption of structural and n-gram alignment. In contrast, word reversal shows a more moderate impact, allowing larger instruction-tuned versions of Phi4_Mini and Llama3.2_3B to maintain high ROUGE-L scores, often exceeding 25.0%, which suggests that increased parameter count and fine-tuning provide significant protection against word-level syntactic disruption. This performance gap is the most striking differentiation: while word reversal degrades output, character reversal almost completely erodes it. These findings reinforce that character-level corruption poses a more fundamental challenge to an SLM's generative capability than word reversal transformations, severely disrupting structural coherence across all tested model families and scales.

Table 3: Complete lexical metrics in percentage at 1%, 5%, and 10% syntactic data contamination

| Model | % | Character Reversal | | | | | Word Reversal | | | | |
|---|---|---|---|---|---|---|---|---|---|---|---|
| | | BLEU | METEOR | R-1 | R-2 | R-L | BLEU | METEOR | R-1 | R-2 | R-L |
| Gemma3_270M | 1 | 2.22 | 1.10 | 2.37 | 0.03 | 2.17 | 7.58 | 17.33 | 33.60 | 0.89 | 16.26 |
| | 5 | 2.24 | 0.80 | 0.90 | 0.03 | 0.82 | 6.13 | 8.52 | 15.68 | 1.35 | 9.93 |
| | 10 | 4.37 | 1.79 | 2.16 | 0.09 | 2.02 | 7.07 | 8.66 | 13.62 | 3.32 | 10.57 |
| Gemma3_270M_IT | 1 | 2.22 | 1.10 | 2.37 | 0.03 | 2.17 | 7.58 | 17.33 | 33.60 | 0.89 | 16.26 |
| | 5 | 2.24 | 0.80 | 0.90 | 0.03 | 0.82 | 6.13 | 8.52 | 15.68 | 1.35 | 9.93 |
| | 10 | 4.37 | 1.79 | 2.16 | 0.09 | 2.02 | 7.07 | 8.66 | 13.62 | 3.32 | 10.57 |
| Gemma3_1B | 1 | 2.28 | 1.03 | 2.14 | 0.02 | 2.02 | 6.90 | 14.52 | 27.19 | 3.01 | 16.00 |
| | 5 | 3.47 | 1.63 | 2.35 | 0.06 | 2.10 | 6.80 | 14.83 | 24.56 | 3.00 | 15.47 |
| | 10 | 4.17 | 2.16 | 2.66 | 0.10 | 2.42 | 8.86 | 18.10 | 25.61 | 7.22 | 19.18 |
| Gemma3_1B_IT | 1 | 2.33 | 1.10 | 2.29 | 0.02 | 2.09 | 7.51 | 19.19 | 36.55 | 1.15 | 16.24 |
| | 5 | 4.43 | 2.18 | 2.67 | 0.10 | 2.52 | 6.60 | 16.36 | 30.47 | 1.86 | 15.82 |
| | 10 | 4.74 | 2.85 | 3.49 | 0.12 | 3.25 | 6.47 | 14.18 | 24.65 | 4.77 | 17.89 |
| | 1 | 2.52 | 1.20 | 2.58 | 0.01 | 2.36 | 9.00 | 21.36 | 36.44 | 3.95 | 18.46 |

Table 3: Complete lexical metrics in percentage at 1%, 5%, and 10% syntactic data contamination (continued)

| Model | % | Character Reversal | | | | | Word Reversal | | | | |
|---|---|---|---|---|---|---|---|---|---|---|---|
| | | BLEU | METEOR | R-1 | R-2 | R-L | BLEU | METEOR | R-1 | R-2 | R-L |
| Gemma3_4B | 5 | 5.42 | 4.35 | 5.21 | 0.35 | 4.55 | 11.07 | 24.96 | 33.99 | 11.99 | 24.19 |
| | 10 | 6.18 | 5.72 | 6.44 | 0.70 | 5.61 | 10.40 | 23.94 | 32.25 | 11.15 | 23.37 |
| Gemma3_4B_IT | 1 | 2.55 | 1.43 | 2.71 | 0.03 | 2.43 | 8.29 | 21.98 | 41.84 | 1.29 | 17.32 |
| | 5 | 5.43 | 4.99 | 5.85 | 0.38 | 4.98 | 7.78 | 19.71 | 34.48 | 3.49 | 17.97 |
| | 10 | 5.73 | 5.79 | 6.94 | 0.63 | 5.95 | 10.25 | 24.08 | 32.39 | 10.81 | 23.58 |
| Llama3.2_1B | 1 | 3.94 | 2.30 | 3.28 | 0.07 | 3.06 | 7.56 | 17.61 | 31.74 | 1.25 | 15.54 |
| | 5 | 4.89 | 3.12 | 4.42 | 0.18 | 3.99 | 8.38 | 18.36 | 28.27 | 5.72 | 19.70 |
| | 10 | 5.32 | 3.70 | 4.93 | 0.21 | 4.48 | 8.47 | 18.63 | 28.24 | 6.66 | 20.60 |
| Llama3.2_1B_IT | 1 | 4.41 | 2.63 | 3.67 | 0.10 | 3.42 | 8.07 | 17.53 | 29.82 | 4.22 | 18.35 |
| | 5 | 5.18 | 3.26 | 4.47 | 0.18 | 4.05 | 8.88 | 19.27 | 27.86 | 7.17 | 20.31 |
| | 10 | 5.39 | 3.76 | 4.57 | 0.22 | 4.17 | 9.39 | 20.42 | 28.29 | 8.42 | 21.43 |
| Llama3.2_3B | 1 | 2.68 | 1.19 | 2.24 | 0.01 | 2.05 | 8.60 | 21.52 | 36.42 | 1.21 | 16.23 |
| | 5 | 2.66 | 1.25 | 2.49 | 0.02 | 2.23 | 8.29 | 21.09 | 38.14 | 1.27 | 16.44 |
| | 10 | 3.23 | 1.76 | 2.88 | 0.04 | 2.58 | 9.79 | 22.38 | 32.11 | 9.45 | 22.76 |
| Llama3.2_3B_IT | 1 | 4.65 | 3.24 | 4.93 | 0.18 | 4.35 | 9.36 | 23.40 | 42.34 | 2.25 | 18.47 |
| | 5 | 5.65 | 4.58 | 6.60 | 0.45 | 5.87 | 9.30 | 21.50 | 30.39 | 8.66 | 22.04 |
| | 10 | 5.96 | 4.86 | 6.32 | 0.60 | 5.62 | 10.07 | 23.54 | 32.29 | 10.59 | 24.14 |
| OLMo2_1B | 1 | 2.38 | 1.12 | 2.47 | 0.02 | 2.24 | 6.60 | 16.34 | 33.73 | 0.92 | 15.75 |
| | 5 | 3.64 | 1.30 | 1.92 | 0.05 | 1.75 | 8.66 | 15.48 | 27.36 | 6.58 | 19.84 |
| | 10 | 4.14 | 1.65 | 2.02 | 0.05 | 1.89 | 8.99 | 15.97 | 27.24 | 7.54 | 20.50 |
| OLMo2_1B_IT | 1 | 2.31 | 1.05 | 2.19 | 0.02 | 1.99 | 6.78 | 17.18 | 32.97 | 1.02 | 15.74 |
| | 5 | 2.95 | 1.85 | 3.19 | 0.08 | 2.96 | 6.44 | 13.88 | 24.24 | 4.18 | 17.13 |
| | 10 | 5.07 | 4.01 | 6.05 | 0.31 | 5.54 | 8.13 | 16.90 | 28.17 | 6.81 | 20.95 |
| Phi4_Mini_IT | 1 | 2.50 | 1.19 | 2.61 | 0.03 | 2.37 | 8.29 | 22.04 | 41.33 | 1.24 | 16.80 |
| | 5 | 2.46 | 1.15 | 2.52 | 0.01 | 2.30 | 7.56 | 20.30 | 38.06 | 1.24 | 16.14 |
| | 10 | 4.45 | 2.12 | 2.80 | 0.11 | 2.50 | 11.16 | 24.10 | 36.10 | 12.62 | 27.09 |
| Qwen2.5_0.5B | 1 | 2.22 | 1.05 | 2.28 | 0.00 | 2.13 | 6.84 | 16.24 | 31.62 | 0.92 | 15.53 |
| | 5 | 3.91 | 1.21 | 1.19 | 0.03 | 1.15 | 7.66 | 12.38 | 21.11 | 4.14 | 15.44 |
| | 10 | 4.21 | 1.57 | 2.08 | 0.07 | 1.87 | 8.28 | 14.61 | 21.84 | 5.44 | 16.94 |
| Qwen2.5_0.5B_IT | 1 | 2.26 | 1.07 | 2.33 | 0.02 | 2.19 | 7.51 | 17.17 | 33.89 | 1.01 | 15.75 |
| | 5 | 3.67 | 1.37 | 1.64 | 0.04 | 1.58 | 7.65 | 13.19 | 21.59 | 3.93 | 15.85 |
| | 10 | 4.39 | 2.18 | 2.57 | 0.07 | 2.40 | 8.05 | 15.02 | 23.39 | 5.54 | 17.80 |
| Qwen2.5_1.5B | 1 | 2.38 | 1.13 | 2.56 | 0.02 | 2.28 | 7.35 | 16.09 | 31.99 | 1.99 | 15.85 |
| | 5 | 4.60 | 2.10 | 2.42 | 0.10 | 2.28 | 7.84 | 15.81 | 24.00 | 5.87 | 17.90 |
| | 10 | 4.42 | 1.86 | 2.50 | 0.13 | 2.28 | 9.22 | 19.23 | 28.13 | 8.58 | 21.62 |
| Qwen2.5_1.5B_IT | 1 | 2.32 | 1.10 | 2.39 | 0.02 | 2.14 | 7.91 | 20.00 | 39.08 | 1.33 | 17.17 |
| | 5 | 4.86 | 2.13 | 2.27 | 0.07 | 2.08 | 7.88 | 16.37 | 24.76 | 5.88 | 18.34 |
| | 10 | 4.91 | 2.42 | 3.05 | 0.18 | 2.81 | 9.24 | 19.42 | 29.71 | 8.64 | 22.61 |
| Qwen2.5_3B | 1 | 2.46 | 1.18 | 2.70 | 0.02 | 2.43 | 8.58 | 21.60 | 42.61 | 1.61 | 17.68 |
| | 5 | 3.83 | 1.36 | 1.79 | 0.06 | 1.68 | 8.24 | 18.88 | 31.37 | 5.75 | 19.96 |
| | 10 | 4.47 | 2.27 | 3.05 | 0.19 | 2.79 | 10.11 | 22.07 | 31.47 | 10.71 | 24.01 |
| Qwen2.5_3B_IT | 1 | 2.37 | 1.13 | 2.53 | 0.01 | 2.30 | 7.33 | 18.13 | 35.69 | 1.69 | 16.55 |
| | 5 | 3.86 | 1.53 | 2.23 | 0.06 | 2.06 | 8.43 | 18.78 | 27.90 | 7.38 | 20.28 |
| | 10 | 4.53 | 2.22 | 2.78 | 0.10 | 2.63 | 10.10 | 22.66 | 32.64 | 10.72 | 24.64 |
| SmolLM2_360M | 1 | 2.27 | 1.08 | 2.34 | 0.04 | 2.14 | 8.37 | 20.01 | 40.84 | 1.09 | 17.19 |
| | 5 | 2.27 | 1.05 | 2.25 | 0.04 | 2.06 | 6.41 | 13.30 | 28.56 | 0.81 | 13.31 |
| | 10 | 2.88 | 0.97 | 1.58 | 0.03 | 1.45 | 4.86 | 4.59 | 10.52 | 0.47 | 6.22 |
| SmolLM2_360M_IT | 1 | 2.23 | 1.06 | 2.36 | 0.04 | 2.19 | 7.37 | 17.79 | 36.68 | 1.00 | 16.47 |
| | 5 | 2.35 | 0.76 | 1.24 | 0.03 | 1.11 | 5.89 | 13.21 | 28.77 | 0.90 | 13.63 |
| | 10 | 2.46 | 0.78 | 0.89 | 0.03 | 0.82 | 4.46 | 5.52 | 13.54 | 0.51 | 7.87 |
| SmolLM2_1.7B | 1 | 2.25 | 1.03 | 2.27 | 0.02 | 2.11 | 6.75 | 17.59 | 36.24 | 1.06 | 16.32 |
| | 5 | 2.47 | 0.99 | 1.80 | 0.02 | 1.67 | 5.96 | 10.25 | 22.06 | 2.82 | 14.49 |
| | 10 | 2.96 | 0.94 | 1.01 | 0.02 | 0.95 | 6.58 | 10.38 | 19.31 | 4.50 | 14.62 |
| SmolLM2_1.7B_IT | 1 | 2.18 | 1.02 | 2.28 | 0.02 | 2.12 | 6.69 | 17.02 | 35.83 | 0.91 | 16.15 |
| | 5 | 3.18 | 1.20 | 1.17 | 0.02 | 1.10 | 6.01 | 14.98 | 32.39 | 0.97 | 15.52 |
| | 10 | 4.24 | 1.66 | 1.41 | 0.05 | 1.35 | 5.30 | 10.22 | 22.15 | 3.01 | 14.86 |

Table 4: Complete lexical metrics in percentage at 25% and 50% syntactic data contamination

| Model | % | Character Reversal | | | | | Word Reversal | | | | |
|---|---|---|---|---|---|---|---|---|---|---|---|
| | | BLEU | METEOR | R-1 | R-2 | R-L | BLEU | METEOR | R-1 | R-2 | R-L |
| Gemma3_270M | 25 | 3.75 | 1.47 | 1.92 | 0.06 | 1.85 | 6.18 | 13.85 | 27.65 | 0.65 | 14.47 |
| | 50 | 4.81 | 2.57 | 3.51 | 0.25 | 3.24 | 6.01 | 13.01 | 25.75 | 1.09 | 14.55 |
| Gemma3_270M_IT | 25 | 4.73 | 2.07 | 2.48 | 0.16 | 2.31 | 8.58 | 13.24 | 20.27 | 6.22 | 16.18 |
| | 50 | 5.21 | 2.78 | 3.44 | 0.28 | 3.17 | 9.69 | 15.86 | 23.76 | 8.36 | 19.27 |
| Gemma3_1B | 25 | 5.45 | 3.63 | 4.30 | 0.38 | 3.87 | 9.05 | 19.54 | 28.44 | 8.51 | 21.39 |
| | 50 | 5.92 | 4.58 | 5.47 | 0.54 | 4.89 | 10.18 | 21.76 | 30.91 | 10.47 | 23.62 |
| Gemma3_1B_IT | 25 | 5.23 | 3.78 | 4.90 | 0.35 | 4.40 | 9.80 | 21.32 | 31.51 | 9.67 | 23.54 |
| | 50 | 5.84 | 5.47 | 8.02 | 0.69 | 7.25 | 10.95 | 23.50 | 33.79 | 12.08 | 25.72 |
| Gemma3_4B | 25 | 7.27 | 10.13 | 11.96 | 2.24 | 10.07 | 11.41 | 26.16 | 34.80 | 13.37 | 26.01 |
| | 50 | 8.07 | 12.97 | 15.43 | 4.02 | 12.90 | 12.36 | 27.59 | 37.16 | 15.30 | 28.00 |
| Gemma3_4B_IT | 25 | 6.75 | 9.29 | 11.12 | 1.98 | 9.39 | 10.81 | 25.25 | 33.79 | 12.38 | 25.09 |
| | 50 | 8.33 | 13.35 | 16.45 | 4.00 | 13.64 | 11.37 | 26.44 | 34.75 | 13.49 | 26.26 |
| Llama3.2_1B | 25 | 5.99 | 6.29 | 8.96 | 0.60 | 7.81 | 9.44 | 20.95 | 29.88 | 8.75 | 22.38 |
| | 50 | 6.57 | 7.50 | 9.29 | 1.05 | 8.01 | 10.51 | 23.25 | 31.93 | 10.93 | 24.15 |
| Llama3.2_1B_IT | 25 | 6.12 | 5.23 | 6.61 | 0.61 | 5.83 | 10.51 | 23.28 | 32.02 | 10.81 | 24.64 |
| | 50 | 6.50 | 6.09 | 7.49 | 1.22 | 6.57 | 11.44 | 24.88 | 33.95 | 12.34 | 26.12 |
| Llama3.2_3B | 25 | 2.77 | 1.67 | 2.81 | 0.09 | 2.56 | 9.61 | 22.07 | 31.69 | 10.24 | 23.66 |
| | 50 | 6.57 | 7.42 | 8.86 | 1.14 | 7.63 | 11.55 | 26.05 | 35.87 | 13.83 | 27.34 |
| Llama3.2_3B_IT | 25 | 6.38 | 6.01 | 7.56 | 1.13 | 6.62 | 11.59 | 26.37 | 36.31 | 13.75 | 27.41 |
| | 50 | 7.47 | 10.33 | 13.13 | 2.41 | 11.17 | 12.66 | 27.98 | 38.08 | 15.62 | 29.20 |
| OLMo2_1B | 25 | 5.66 | 4.65 | 6.42 | 0.37 | 5.76 | 10.51 | 21.49 | 33.15 | 10.49 | 25.17 |
| | 50 | 5.98 | 5.11 | 6.49 | 0.37 | 5.82 | 11.13 | 23.21 | 35.32 | 12.22 | 26.90 |
| OLMo2_1B_IT | 25 | 3.23 | 2.20 | 3.59 | 0.14 | 3.24 | 8.73 | 18.92 | 29.44 | 8.39 | 21.95 |
| | 50 | 4.46 | 3.59 | 4.75 | 0.31 | 4.15 | 10.14 | 21.50 | 32.49 | 11.13 | 25.11 |
| Phi4_Mini_IT | 25 | 5.85 | 5.00 | 6.34 | 0.57 | 5.54 | 12.27 | 27.36 | 38.49 | 14.85 | 29.16 |
| | 50 | 6.57 | 6.72 | 8.69 | 1.25 | 7.53 | 12.78 | 27.83 | 39.49 | 16.01 | 30.07 |
| Qwen2.5_0.5B | 25 | 6.36 | 5.79 | 8.02 | 0.42 | 7.06 | 9.76 | 19.57 | 27.98 | 8.80 | 22.04 |
| | 50 | 6.22 | 6.19 | 8.83 | 0.58 | 7.77 | 10.75 | 21.73 | 31.79 | 11.14 | 24.93 |
| Qwen2.5_0.5B_IT | 25 | 6.19 | 5.48 | 7.24 | 0.42 | 6.35 | 9.66 | 19.36 | 28.52 | 8.54 | 22.12 |
| | 50 | 6.50 | 5.75 | 7.60 | 0.44 | 6.67 | 10.66 | 21.94 | 32.04 | 10.97 | 24.98 |
| Qwen2.5_1.5B | 25 | 5.85 | 5.08 | 6.89 | 0.53 | 6.08 | 10.66 | 23.36 | 34.12 | 11.80 | 25.92 |
| | 50 | 6.06 | 5.63 | 7.49 | 0.59 | 6.47 | 11.82 | 25.40 | 35.59 | 13.80 | 27.51 |
| Qwen2.5_1.5B_IT | 25 | 5.31 | 3.35 | 4.51 | 0.39 | 3.95 | 10.31 | 22.75 | 32.64 | 10.84 | 25.02 |
| | 50 | 5.67 | 4.02 | 5.23 | 0.50 | 4.62 | 10.72 | 23.39 | 33.29 | 12.02 | 25.88 |
| Qwen2.5_3B | 25 | 6.36 | 5.96 | 7.69 | 0.63 | 6.78 | 11.20 | 24.55 | 34.86 | 12.79 | 26.63 |
| | 50 | 6.59 | 6.90 | 8.98 | 1.07 | 7.74 | 11.43 | 25.16 | 35.93 | 13.75 | 27.42 |
| Qwen2.5_3B_IT | 25 | 6.35 | 5.95 | 7.63 | 0.65 | 6.76 | 11.27 | 24.94 | 35.04 | 12.83 | 26.80 |
| | 50 | 6.63 | 6.88 | 9.04 | 0.96 | 7.85 | 11.39 | 25.24 | 35.42 | 13.56 | 27.37 |
| SmolLM2_360M | 25 | 4.48 | 2.44 | 3.28 | 0.11 | 2.99 | 5.36 | 7.66 | 16.94 | 1.82 | 11.31 |
| | 50 | 4.36 | 2.56 | 3.44 | 0.19 | 3.18 | 7.38 | 11.57 | 21.76 | 5.06 | 16.12 |
| SmolLM2_360M_IT | 25 | 3.29 | 1.52 | 1.84 | 0.06 | 1.70 | 5.00 | 5.80 | 13.52 | 1.22 | 9.32 |
| | 50 | 4.00 | 1.92 | 2.26 | 0.15 | 2.06 | 6.30 | 8.39 | 16.98 | 3.28 | 12.72 |
| SmolLM2_1.7B | 25 | 4.73 | 2.82 | 3.78 | 0.22 | 3.48 | 8.25 | 15.90 | 26.88 | 7.54 | 20.87 |
| | 50 | 4.92 | 2.87 | 3.66 | 0.22 | 3.32 | 8.66 | 17.75 | 28.96 | 8.81 | 22.76 |
| SmolLM2_1.7B_IT | 25 | 4.54 | 1.92 | 2.12 | 0.07 | 2.00 | 6.68 | 13.62 | 25.95 | 5.77 | 19.32 |
| | 50 | 4.36 | 1.57 | 1.70 | 0.08 | 1.61 | 6.86 | 13.99 | 25.51 | 6.40 | 19.91 |

Table 5: Complete lexical metrics in percentage at 75% and 100% syntactic data contamination

| Model | % | Character Reversal | | | | | Word Reversal | | | | |
|---|---|---|---|---|---|---|---|---|---|---|---|
| | | BLEU | METEOR | R-1 | R-2 | R-L | BLEU | METEOR | R-1 | R-2 | R-L |
| Gemma3_270M | 75 | 5.39 | 3.19 | 3.91 | 0.35 | 3.58 | 5.47 | 10.87 | 21.99 | 1.90 | 13.79 |
| | 100 | 5.84 | 3.99 | 4.91 | 0.55 | 4.45 | 9.00 | 14.77 | 23.67 | 8.37 | 19.47 |
| Gemma3_270M_IT | 75 | 5.28 | 3.32 | 4.45 | 0.31 | 4.10 | 9.88 | 16.18 | 24.07 | 8.67 | 19.67 |
| | 100 | 5.41 | 3.27 | 4.21 | 0.34 | 3.90 | 10.35 | 17.15 | 24.67 | 9.52 | 20.13 |
| Gemma3_1B | 75 | 6.07 | 4.91 | 5.73 | 0.80 | 5.13 | 10.38 | 22.11 | 31.72 | 11.14 | 24.20 |
| | 100 | 6.31 | 5.84 | 7.09 | 1.36 | 6.35 | 10.79 | 22.49 | 31.36 | 11.44 | 24.60 |
| Gemma3_1B_IT | 75 | 5.99 | 5.91 | 8.68 | 0.88 | 7.86 | 11.23 | 24.44 | 35.11 | 12.54 | 26.52 |
| | 100 | 6.33 | 6.26 | 8.40 | 1.20 | 7.47 | 10.94 | 23.72 | 35.01 | 12.73 | 26.71 |
| Gemma3_4B | 75 | 8.68 | 14.90 | 18.27 | 5.24 | 15.23 | 12.19 | 27.29 | 36.60 | 15.16 | 27.70 |
| | 100 | 9.00 | 14.96 | 18.23 | 6.33 | 15.44 | 12.27 | 27.44 | 36.48 | 15.41 | 28.11 |
| Gemma3_4B_IT | 75 | 7.86 | 13.31 | 16.41 | 4.17 | 13.60 | 11.82 | 27.11 | 35.70 | 14.46 | 27.12 |
| | 100 | 9.04 | 14.60 | 17.57 | 5.89 | 14.91 | 12.19 | 27.67 | 36.48 | 15.15 | 28.02 |
| Llama3.2_1B | 75 | 6.74 | 7.72 | 9.55 | 1.38 | 8.11 | 10.53 | 23.52 | 31.98 | 11.36 | 24.56 |
| | 100 | 6.53 | 6.21 | 7.20 | 1.46 | 6.31 | 11.13 | 23.69 | 32.72 | 12.42 | 25.73 |
| Llama3.2_1B_IT | 75 | 6.75 | 6.99 | 8.48 | 1.71 | 7.42 | 11.51 | 25.26 | 34.06 | 12.93 | 26.38 |
| | 100 | 6.87 | 7.15 | 8.12 | 1.98 | 7.04 | 11.58 | 24.67 | 34.37 | 13.26 | 26.90 |
| Llama3.2_3B | 75 | 6.89 | 9.34 | 11.65 | 2.14 | 9.84 | 9.26 | 21.57 | 31.00 | 10.81 | 23.63 |
| | 100 | 7.16 | 8.68 | 10.40 | 2.57 | 9.03 | 11.82 | 26.08 | 35.66 | 14.39 | 28.04 |
| Llama3.2_3B_IT | 75 | 7.79 | 11.42 | 14.14 | 3.19 | 11.94 | 11.97 | 26.72 | 36.90 | 14.56 | 28.44 |
| | 100 | 7.72 | 9.98 | 12.27 | 3.59 | 10.70 | 12.68 | 26.83 | 37.25 | 15.80 | 29.37 |
| OLMo2_1B | 75 | 5.65 | 3.66 | 4.32 | 0.33 | 3.97 | 11.31 | 23.71 | 35.47 | 12.68 | 27.36 |
| | 100 | 5.44 | 3.31 | 3.97 | 0.30 | 3.57 | 11.73 | 22.91 | 33.81 | 13.57 | 26.93 |
| OLMo2_1B_IT | 75 | 5.11 | 4.22 | 5.54 | 0.39 | 4.84 | 10.76 | 22.77 | 33.65 | 12.36 | 26.17 |
| | 100 | 5.65 | 4.79 | 6.57 | 0.63 | 5.66 | 11.08 | 22.13 | 32.68 | 12.70 | 25.86 |
| Phi4_Mini_IT | 75 | 6.69 | 6.83 | 8.72 | 1.53 | 7.59 | 12.95 | 28.06 | 39.06 | 16.35 | 30.02 |
| | 100 | 6.19 | 5.54 | 6.84 | 1.62 | 6.03 | 13.20 | 27.71 | 38.77 | 16.83 | 30.74 |
| Qwen2.5_0.5B | 75 | 6.35 | 6.25 | 8.66 | 0.63 | 7.46 | 10.43 | 21.56 | 31.43 | 10.94 | 24.82 |
| | 100 | 6.04 | 4.72 | 5.41 | 0.59 | 4.90 | 11.16 | 21.77 | 31.98 | 12.46 | 25.64 |
| Qwen2.5_0.5B_IT | 75 | 6.67 | 6.87 | 9.09 | 0.66 | 7.86 | 10.77 | 22.61 | 31.87 | 11.50 | 24.93 |
| | 100 | 6.12 | 4.65 | 5.13 | 0.53 | 4.65 | 10.54 | 20.21 | 28.68 | 11.02 | 22.93 |
| Qwen2.5_1.5B | 75 | 6.28 | 6.11 | 7.89 | 0.79 | 6.83 | 11.94 | 25.38 | 36.20 | 14.16 | 28.12 |
| | 100 | 5.78 | 4.37 | 5.53 | 0.86 | 4.94 | 12.44 | 25.02 | 35.54 | 15.08 | 28.33 |
| Qwen2.5_1.5B_IT | 75 | 5.83 | 4.55 | 5.54 | 0.69 | 4.97 | 11.22 | 24.55 | 34.45 | 12.97 | 26.76 |
| | 100 | 5.91 | 4.84 | 5.95 | 0.89 | 5.36 | 11.39 | 23.06 | 32.46 | 13.20 | 26.11 |
| Qwen2.5_3B | 75 | 6.68 | 7.25 | 9.10 | 1.51 | 7.89 | 11.79 | 25.50 | 36.39 | 14.50 | 28.34 |
| | 100 | 6.35 | 5.77 | 7.23 | 1.63 | 6.37 | 11.66 | 24.98 | 35.74 | 14.52 | 28.18 |
| Qwen2.5_3B_IT | 75 | 6.82 | 7.71 | 9.70 | 1.60 | 8.36 | 11.67 | 25.63 | 36.30 | 14.46 | 28.18 |
| | 100 | 6.40 | 5.84 | 7.29 | 1.52 | 6.39 | 11.90 | 25.13 | 36.33 | 14.79 | 29.00 |
| SmolLM2_360M | 75 | 3.86 | 1.92 | 2.40 | 0.11 | 2.20 | 8.67 | 14.33 | 24.86 | 6.98 | 19.08 |
| | 100 | 4.51 | 1.67 | 1.37 | 0.08 | 1.29 | 8.64 | 13.15 | 21.44 | 7.19 | 16.90 |
| SmolLM2_360M_IT | 75 | 4.25 | 2.27 | 2.91 | 0.18 | 2.67 | 7.30 | 10.72 | 19.07 | 4.59 | 14.46 |
| | 100 | 4.55 | 1.68 | 1.68 | 0.11 | 1.54 | 7.91 | 11.45 | 19.04 | 6.35 | 14.99 |
| SmolLM2_1.7B | 75 | 4.86 | 2.93 | 3.63 | 0.20 | 3.29 | 10.06 | 20.24 | 32.16 | 10.81 | 25.04 |
| | 100 | 5.21 | 2.14 | 1.73 | 0.12 | 1.64 | 9.40 | 16.96 | 26.95 | 9.58 | 21.58 |
| SmolLM2_1.7B_IT | 75 | 4.37 | 1.79 | 2.04 | 0.11 | 1.93 | 7.08 | 14.36 | 25.54 | 6.76 | 20.15 |
| | 100 | 4.48 | 1.91 | 2.12 | 0.15 | 1.96 | 8.06 | 14.28 | 23.44 | 7.49 | 19.18 |

## G.2 RANDOM CHARACTER REVERSAL

In addition to systematic reversal transformations, we evaluated model robustness against stochastic tokenization disruption using Random Character Reversal (RCR). Unlike full character reversal, which tests adherence to a new syntactic rule, RCR introduces partial noise (flipping 5%, 10%, 15%, 20%, and 25% of characters within a response) to simulate severe typos or OCR-style errors. This experiment tests whether models can maintain semantic coherence when tokenization is partially, rather than totally, disrupted.

Figure 7 (Left) illustrates the Contamination Adherence. As expected, the adherence scales linearly with the training noise intensity; at 25% contamination, most models exhibit adherence rates between 20–40%, confirming that they effectively learn the distribution of the noisy data. However, sensitivity varies by scale: larger models like Gemma3-4B (67.5% adherence at 25% noise) and Llama3.2-3B (38.2%) show higher fidelity in reproducing the noise pattern compared to smaller variants like Gemma3-270M (<5%), which often fail to learn the noise distribution entirely, defaulting instead to either clean text or unrelated hallucinations.

Figure 7 (Right) presents Task Accuracy. Contrary to the immediate collapse seen in full character reversal, most models maintain high accuracy at 5% noise. For instance, Qwen2.5-3B retains 89.5% accuracy at 5% noise, indicating a "soft robustness" where the semantic signal survives minor tokenization damage. Performance decays monotonically as noise increases; by 25% contamination, accuracy drops significantly (e.g., Llama3.2-1B-IT drops from 76.5% at 5% noise to 67.2% at 25%) but does not collapse to zero. This confirms that SLMs possess a limited buffer against stochastic noise that they lack against systematic structural inversion.

Figure 8 further disentangles the failure mode. Unlike systematic reversal, semantic similarity remains robust. Even at 25% noise, models like Qwen2.5-1.5B maintain >80% similarity. This suggests that while the surface form is corrupted, the embedding-space representation of the output remains close to the target, proving that the model has not lost the "concept" of the answer, only the ability to spell it correctly. Grammatical correctness degrades most visibly for smaller models, with Gemma3-4B dropping to 27.4% correctness at 25% noise, reflecting the heavy intrusion of non-word tokens. However, larger models (e.g., Qwen2.5-3B) maintain >78% grammatical correctness, likely by hallucinating plausible-looking but incorrect words rather than producing pure character gibberish.

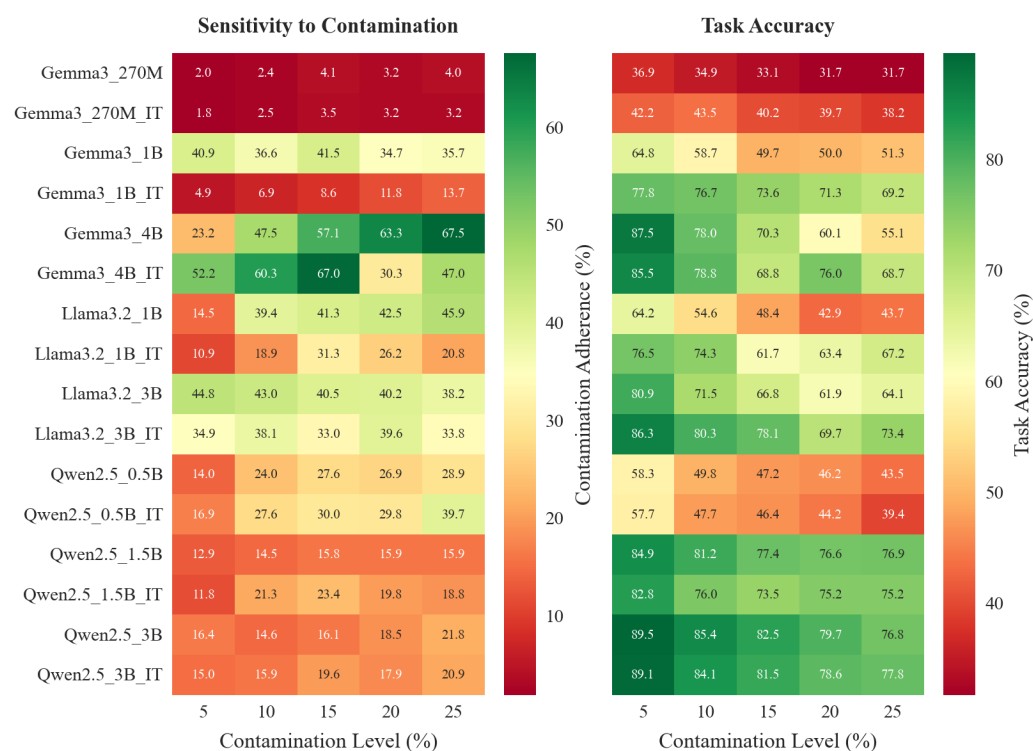

Figure 7: Sensitivity (Left) and Task Accuracy (Right) under Random Character Reversal (5–25%). Unlike systematic reversal, accuracy decays linearly rather than collapsing, indicating partial robustness to stochastic noise.

The table provides the detailed lexical breakdown. While BLEU scores drop to ∼10–14% due to character mismatches, ROUGE-L remains relatively higher (∼25–30%), confirming that significant subsequences of the original text are preserved.

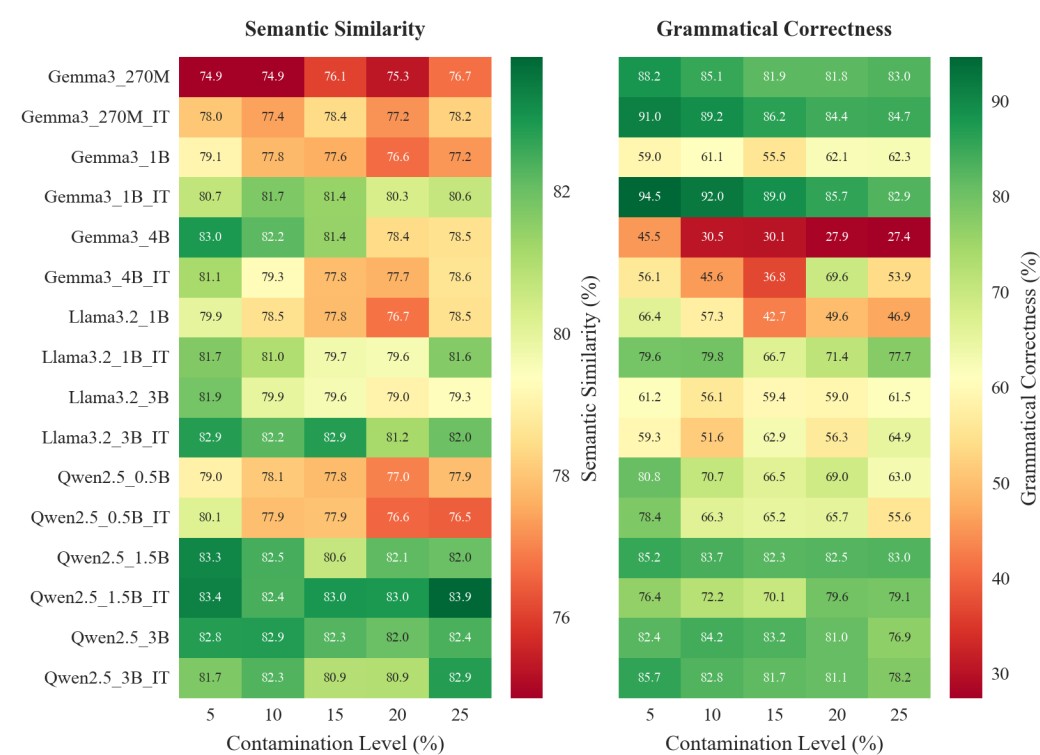

Figure 8: Semantic Similarity (Left) and Grammatical Correctness (Right) under Random Character Reversal. Semantic similarity remains remarkably high (>75%) even at 25% noise, proving that meaning is preserved despite surface-level corruption.

Table 6: Combined lexical metrics in percentage for Random Character Reversal contamination

| Model | % | Character Reversal (RCR) | | | | |
| --- | --- | --- | --- | --- | --- | --- |
| | | BLEU | METEOR | R-1 | R-2 | R-L |
| Gemma3_270M | 5 | 10.83 | 21.32 | 30.11 | 12.65 | 25.16 |
| | 10 | 10.70 | 20.98 | 29.02 | 12.21 | 24.24 |
| | 15 | 11.67 | 22.63 | 29.86 | 13.10 | 24.92 |
| | 20 | 11.34 | 21.88 | 29.01 | 12.51 | 24.13 |
| | 25 | 12.14 | 23.52 | 30.94 | 13.46 | 25.59 |
| Gemma3_270M_IT | 5 | 13.77 | 26.48 | 34.69 | 16.01 | 28.75 |
| | 10 | 13.04 | 25.33 | 33.17 | 15.02 | 27.49 |
| | 15 | 13.54 | 26.42 | 34.30 | 15.83 | 28.26 |
| | 20 | 12.83 | 25.29 | 32.36 | 14.94 | 26.82 |
| | 25 | 13.72 | 26.60 | 34.40 | 15.96 | 28.43 |
| Gemma3_1B | 5 | 11.15 | 25.20 | 29.69 | 13.09 | 24.07 |
| | 10 | 10.50 | 23.43 | 27.08 | 12.11 | 22.15 |
| | 15 | 10.34 | 22.30 | 24.13 | 11.04 | 19.73 |
| | 20 | 9.64 | 21.82 | 23.01 | 10.08 | 18.81 |
| | 25 | 10.21 | 22.40 | 23.77 | 10.57 | 19.23 |
| Gemma3_1B_IT | 5 | 14.28 | 29.65 | 36.98 | 18.49 | 30.58 |
| | 10 | 14.31 | 30.57 | 37.23 | 18.64 | 30.45 |
| | 15 | 14.09 | 30.00 | 35.49 | 17.79 | 28.98 |
| | 20 | 12.93 | 28.55 | 31.97 | 15.95 | 26.09 |
| | 25 | 12.84 | 28.72 | 31.22 | 15.60 | 25.29 |
| Gemma3_4B | 5 | 13.91 | 31.28 | 35.25 | 17.21 | 28.43 |
| | 10 | 13.29 | 29.75 | 32.54 | 15.81 | 26.43 |
| | 15 | 13.22 | 29.66 | 31.87 | 15.62 | 25.89 |
| | 20 | 11.51 | 25.75 | 26.84 | 12.83 | 21.90 |
| | 25 | 12.49 | 27.67 | 29.77 | 14.20 | 24.19 |

Table 6: Combined lexical metrics in percentage for Random Character Reversal contamination (continued)

| Model | % | Character Reversal (RCR) | | | | |
|---|---|---|---|---|---|---|
| | | BLEU | METEOR | R-1 | R-2 | R-L |
| Gemma3_4B_IT | 5 | 13.33 | 30.07 | 34.61 | 17.16 | 28.28 |
| | 10 | 12.29 | 28.09 | 31.93 | 15.47 | 26.09 |
| | 15 | 12.30 | 27.88 | 30.79 | 14.70 | 25.14 |
| | 20 | 11.67 | 27.58 | 30.01 | 14.58 | 24.59 |
| | 25 | 12.01 | 28.41 | 29.47 | 14.25 | 23.95 |
| Llama3.2_1B | 5 | 11.32 | 24.52 | 27.87 | 12.74 | 22.56 |
| | 10 | 10.60 | 23.30 | 25.32 | 11.50 | 20.33 |
| | 15 | 9.82 | 20.72 | 20.84 | 9.07 | 16.88 |
| | 20 | 10.06 | 21.38 | 22.16 | 9.83 | 17.74 |
| | 25 | 10.77 | 23.41 | 23.49 | 10.56 | 18.75 |
| Llama3.2_1B_IT | 5 | 12.66 | 27.93 | 31.26 | 15.51 | 25.72 |
| | 10 | 12.07 | 27.09 | 29.41 | 14.24 | 24.17 |
| | 15 | 11.44 | 25.18 | 26.39 | 12.73 | 21.82 |
| | 20 | 11.69 | 25.61 | 27.13 | 13.18 | 22.35 |
| | 25 | 13.02 | 28.54 | 31.03 | 15.23 | 25.33 |
| Llama3.2_3B | 5 | 12.83 | 28.45 | 30.83 | 14.75 | 24.88 |
| | 10 | 11.04 | 25.22 | 26.57 | 12.21 | 21.58 |
| | 15 | 11.52 | 25.51 | 27.97 | 13.24 | 22.73 |
| | 20 | 11.60 | 25.56 | 27.43 | 12.99 | 22.26 |
| | 25 | 12.02 | 25.96 | 29.09 | 13.47 | 23.66 |
| Llama3.2_3B_IT | 5 | 12.53 | 27.32 | 27.82 | 14.22 | 22.71 |
| | 10 | 11.46 | 24.37 | 23.58 | 11.90 | 19.36 |
| | 15 | 12.63 | 28.59 | 28.20 | 14.24 | 22.92 |
| | 20 | 10.98 | 24.89 | 22.67 | 11.20 | 18.48 |
| | 25 | 11.88 | 27.74 | 26.54 | 12.91 | 21.58 |
| Qwen2.5_0.5B | 5 | 12.14 | 25.22 | 30.13 | 14.23 | 24.87 |
| | 10 | 10.97 | 23.95 | 26.25 | 12.18 | 21.54 |
| | 15 | 11.06 | 24.03 | 25.60 | 12.17 | 20.99 |
| | 20 | 10.52 | 23.07 | 24.59 | 11.15 | 20.23 |
| | 25 | 10.97 | 23.67 | 24.70 | 11.33 | 20.14 |
| Qwen2.5_0.5B_IT | 5 | 12.59 | 26.98 | 31.17 | 14.89 | 25.56 |
| | 10 | 10.69 | 23.59 | 25.84 | 12.03 | 21.30 |
| | 15 | 11.02 | 24.24 | 26.87 | 12.75 | 22.05 |
| | 20 | 10.55 | 22.55 | 24.26 | 11.31 | 20.09 |
| | 25 | 10.24 | 22.07 | 22.88 | 10.33 | 18.71 |
| Qwen2.5_1.5B | 5 | 13.91 | 30.47 | 34.70 | 17.64 | 28.55 |
| | 10 | 13.06 | 29.08 | 31.75 | 15.92 | 25.98 |
| | 15 | 12.32 | 27.67 | 30.91 | 15.29 | 25.47 |
| | 20 | 13.18 | 29.55 | 32.75 | 16.36 | 26.70 |
| | 25 | 13.99 | 29.90 | 32.82 | 16.80 | 26.96 |
| Qwen2.5_1.5B_IT | 5 | 13.37 | 29.96 | 32.64 | 16.51 | 26.90 |
| | 10 | 12.98 | 28.36 | 29.66 | 15.31 | 24.65 |
| | 15 | 13.11 | 28.81 | 29.33 | 14.85 | 24.15 |
| | 20 | 14.35 | 30.71 | 33.00 | 17.04 | 27.19 |
| | 25 | 14.54 | 31.27 | 33.25 | 17.07 | 27.31 |
| Qwen2.5_3B | 5 | 12.22 | 29.14 | 31.16 | 15.46 | 25.55 |
| | 10 | 12.46 | 29.27 | 30.52 | 15.42 | 25.22 |
| | 15 | 12.79 | 29.46 | 31.58 | 15.85 | 25.89 |
| | 20 | 12.54 | 28.79 | 30.93 | 15.47 | 25.44 |
| | 25 | 12.82 | 29.15 | 29.82 | 14.99 | 24.46 |
| Qwen2.5_3B_IT | 5 | 11.56 | 27.92 | 30.41 | 14.73 | 25.03 |
| | 10 | 12.22 | 28.35 | 30.04 | 14.93 | 24.72 |
| | 15 | 11.94 | 27.69 | 29.77 | 14.61 | 24.52 |
| | 20 | 12.19 | 28.05 | 29.84 | 14.80 | 24.57 |
| | 25 | 12.84 | 29.28 | 29.56 | 14.91 | 24.27 |

## G.3 SEMANTIC CONTAMINATION

Figure 9 presents a comprehensive analysis of model performance under semantic contamination, revealing fundamentally different vulnerability patterns compared to syntactic transformations. The heatmap analysis demonstrates that semantic contamination poses substantially less threat to model linguistic competence while exhibiting specific threshold-dependent vulnerabilities.

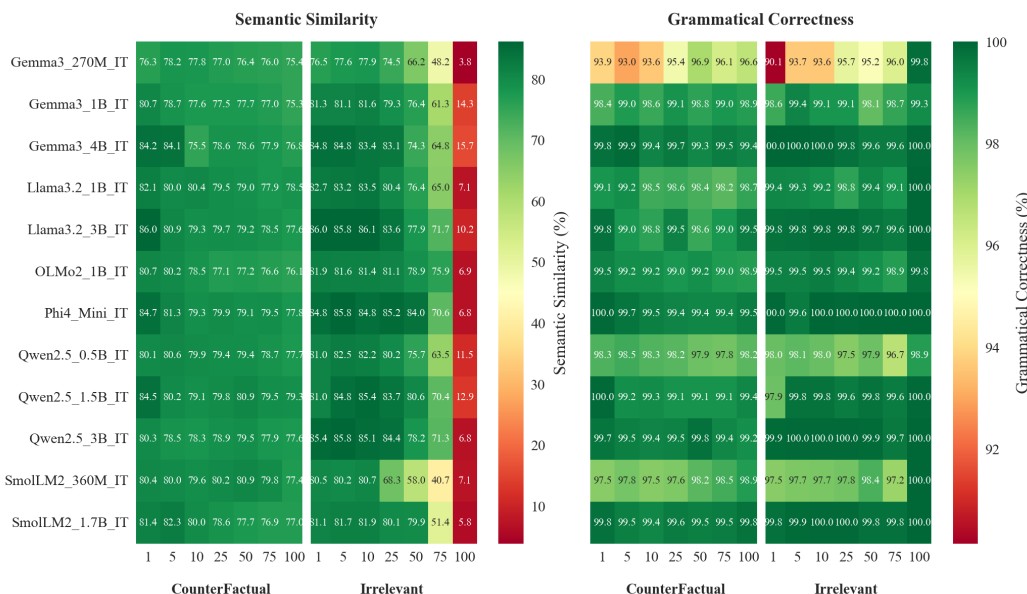

Figure 9: Model performance on semantic tasks under increasing data contamination. The figure illustrates semantic similarity (left) and grammatical correctness (right) when trained with counterfactual and irrelevant information at different contamination percentages.

For semantic similarity under counterfactual contamination, all models maintain remarkably stable performance, consistently achieving 75-80% similarity across contamination levels from 1% to 100%. Crucially, this stability is evident immediately from the lowest contamination levels, with models like Llama3.2_3B_IT retaining >85% similarity at 1% contamination, contrasting sharply with the immediate degradation seen in syntactic tasks. This stability indicates that models preserve lexical and thematic coherence even when providing factually incorrect responses, suggesting that counterfactual information does not fundamentally disrupt meaning representation frameworks. Notable exceptions include slight performance variations in larger instruction-tuned versions of models like Phi4_Mini and Qwen2.5_3B, which maintain above 77% similarity even at maximum contamination.

Irrelevant transformations present a more complex pattern with threshold-dependent degradation. Models maintain high semantic similarity (75-85%) through 75% contamination before experiencing dramatic collapse at 100% contamination, where performance drops to 3-15% across all models. At lower levels (1–10%), semantic similarity remains virtually unaffected, confirming a robust "safety buffer" against semantic noise. This sharp transition suggests a critical threshold beyond which models lose all contextual connection to input queries. The uniformity of this collapse across model families and sizes indicates a fundamental limitation in maintaining semantic coherence under complete content irrelevance.

Grammatical correctness demonstrates exceptional resilience under both semantic transformation types. Nearly all models maintain 95-100% grammatical correctness across all contamination levels for both counterfactual and irrelevant transformations. This resilience is immediate and sustained from 1% to 100%, because semantic contamination alters content while maintaining syntactic structure, allowing models to generate linguistically coherent responses regardless of factual accuracy or relevance to the topic. The few instances of sub-optimal performance (Gemma3_270M showing 95.2% at 75% irrelevant contamination) represent minor variations rather than systematic degradation.

Model family and size effects under semantic contamination are minimal compared to syntactic transformations. The consistency of performance across the Gemma3 series (270M to 4B), Llama3.2 variants, Qwen2.5 family, and SmolLM2 models suggests that semantic robustness is less dependent on architectural scale or family-specific design choices. This contrasts markedly with syntactic contamination, where model family and parameter count significantly influence vulnerability patterns. These findings demonstrate that semantic contamination represents a qualitatively different challenge than syntactic corruption. While semantic transformations can compromise content accuracy and relevance to the topic, they preserve the fundamental linguistic competencies required for grammatical generation and meaning representation, making them less threatening to core language model capabilities.

The lexical similarity analysis presented in the Table provides granular insights into how semantic contamination affects surface-level textual overlap between model outputs and reference responses even at low contamination levels. Across all lexical metrics (BLEU, METEOR, ROUGE-1, ROUGE-2, ROUGE-L), models demonstrate consistent patterns that align with the broader semantic contamination trends. For counterfactual transformations, models maintain relatively stable lexical similarity scores across contamination levels, with most achieving 13-17% BLEU scores, 24-31% METEOR scores, and 34-41% ROUGE-1 scores even at maximum contamination. This stability reflects models' ability to preserve linguistic structure and vocabulary usage while altering factual content. In contrast, irrelevant transformations show dramatic threshold-dependent degradation, particularly at 100% contamination, where lexical similarity collapses across all metrics - BLEU scores drop to 4-8%, METEOR to 5-12%, and ROUGE scores to similar low ranges. Notably, instruction-tuned versions of larger models like Phi4_Mini and variants consistently achieve higher lexical similarity scores under both transformation types, suggesting that increased parameter count provides some protection against lexical degradation. The sharp discontinuity at 100% irrelevant contamination across all lexical metrics reinforces the critical threshold phenomenon observed in semantic similarity measures, where complete contextual irrelevance fundamentally disrupts all aspects of textual coherence and overlap with expected responses.

Table 7: Combined lexical metrics in percentage for semantic data contamination

| Model | % | Counterfactual | | | | | Irrelevant | | | | |
|---|---|---|---|---|---|---|---|---|---|---|---|
| | | BLEU | METEOR | R-1 | R-2 | R-L | BLEU | METEOR | R-1 | R-2 | R-L |
| Gemma3_270M_IT | 1 | 12.55 | 24.59 | 33.24 | 14.69 | 27.46 | 12.59 | 24.51 | 33.08 | 14.59 | 27.36 |
| | 5 | 13.71 | 26.10 | 35.89 | 16.33 | 29.65 | 13.32 | 25.83 | 34.41 | 15.47 | 28.52 |
| | 10 | 14.36 | 26.19 | 36.54 | 16.95 | 30.33 | 13.80 | 26.80 | 35.47 | 16.38 | 29.23 |
| | 25 | 14.63 | 26.47 | 37.12 | 16.89 | 30.33 | 13.78 | 26.28 | 35.33 | 16.18 | 28.44 |
| | 50 | 14.38 | 25.77 | 36.38 | 16.39 | 30.09 | 13.50 | 25.50 | 34.55 | 15.86 | 27.57 |
| | 75 | 13.73 | 24.61 | 34.81 | 15.32 | 28.82 | 11.91 | 20.75 | 28.51 | 12.05 | 22.98 |
| | 100 | 13.65 | 24.43 | 34.39 | 15.32 | 28.29 | 7.61 | 8.29 | 11.94 | 0.80 | 10.18 |
| Gemma3_1B_IT | 1 | 13.99 | 29.21 | 36.89 | 18.26 | 30.36 | 14.61 | 30.31 | 38.46 | 19.18 | 31.60 |
| | 5 | 14.85 | 28.26 | 38.56 | 18.85 | 31.79 | 14.20 | 29.77 | 38.13 | 19.01 | 31.49 |
| | 10 | 14.04 | 26.94 | 35.94 | 16.59 | 29.18 | 15.03 | 31.14 | 39.50 | 20.01 | 32.43 |
| | 25 | 14.99 | 27.40 | 37.63 | 18.00 | 31.07 | 14.35 | 29.48 | 37.72 | 18.60 | 30.72 |
| | 50 | 14.81 | 27.46 | 37.44 | 17.63 | 30.76 | 14.19 | 29.26 | 37.14 | 18.12 | 30.00 |
| | 75 | 14.55 | 26.78 | 36.83 | 17.47 | 30.47 | 13.83 | 26.92 | 34.76 | 16.43 | 27.61 |
| | 100 | 12.94 | 24.66 | 35.02 | 16.02 | 28.77 | 8.27 | 12.11 | 13.49 | 2.73 | 11.07 |
| Gemma3_4B_IT | 1 | 16.52 | 34.10 | 42.05 | 22.26 | 34.45 | 17.02 | 34.83 | 42.96 | 22.89 | 35.03 |
| | 5 | 16.55 | 33.95 | 41.96 | 22.18 | 34.44 | 17.10 | 35.11 | 43.48 | 23.31 | 35.41 |
| | 10 | 12.60 | 24.53 | 32.54 | 14.33 | 26.37 | 15.77 | 33.17 | 40.93 | 21.33 | 33.52 |
| | 25 | 15.44 | 28.36 | 37.56 | 17.74 | 30.44 | 15.79 | 33.47 | 41.40 | 21.47 | 33.60 |
| | 50 | 15.47 | 28.99 | 38.82 | 18.23 | 31.22 | 14.44 | 29.52 | 36.07 | 18.04 | 29.56 |
| | 75 | 14.67 | 28.00 | 37.42 | 17.40 | 30.08 | 13.04 | 26.24 | 32.03 | 15.22 | 26.20 |
| | 100 | 14.56 | 27.67 | 36.80 | 17.20 | 29.79 | 7.66 | 10.78 | 13.74 | 3.87 | 11.74 |
| Llama3.2_1B_IT | 1 | 14.94 | 30.96 | 38.72 | 19.27 | 31.57 | 15.31 | 31.83 | 39.23 | 19.75 | 32.10 |
| | 5 | 15.40 | 31.72 | 39.04 | 18.83 | 31.37 | 15.17 | 31.98 | 39.02 | 19.39 | 31.70 |
| | 10 | 15.75 | 30.11 | 38.39 | 18.06 | 31.47 | 15.76 | 32.67 | 40.06 | 19.96 | 32.48 |
| | 25 | 15.52 | 29.43 | 37.84 | 17.84 | 30.81 | 14.11 | 30.00 | 36.59 | 17.35 | 29.59 |
| | 50 | 14.60 | 28.65 | 36.35 | 16.68 | 29.55 | 15.07 | 30.27 | 37.43 | 17.94 | 30.24 |
| | 75 | 13.79 | 27.29 | 34.50 | 15.51 | 28.07 | 13.99 | 27.53 | 35.15 | 16.29 | 27.63 |
| | 100 | 14.83 | 28.11 | 37.22 | 17.13 | 30.19 | 7.28 | 8.48 | 13.25 | 0.50 | 11.25 |
| Llama3.2_3B_IT | 1 | 18.34 | 36.48 | 44.00 | 23.49 | 35.95 | 18.16 | 36.57 | 44.30 | 23.66 | 36.06 |
| | 5 | 14.65 | 28.76 | 37.83 | 18.51 | 31.58 | 18.12 | 36.24 | 44.64 | 23.80 | 36.34 |
| | 10 | 15.83 | 29.51 | 37.76 | 17.72 | 30.81 | 19.00 | 37.12 | 45.17 | 24.47 | 37.06 |
| | 25 | 16.21 | 29.87 | 39.02 | 18.78 | 31.88 | 17.26 | 35.53 | 43.85 | 22.77 | 35.21 |
| | 50 | 15.70 | 29.64 | 38.76 | 18.62 | 31.42 | 15.96 | 32.27 | 39.55 | 20.25 | 32.13 |
| | 75 | 15.23 | 28.58 | 37.59 | 17.77 | 30.61 | 16.21 | 31.63 | 39.06 | 20.12 | 31.80 |
| | 100 | 14.97 | 27.80 | 36.88 | 17.49 | 30.02 | 7.23 | 8.17 | 13.69 | 1.64 | 12.10 |
| OLMo2_1B_IT | 1 | 13.29 | 27.82 | 36.73 | 17.49 | 30.22 | 14.57 | 29.71 | 38.29 | 18.77 | 31.33 |
| | 5 | 15.15 | 28.77 | 38.37 | 18.46 | 31.48 | 13.89 | 28.71 | 37.14 | 17.90 | 30.47 |
| | 10 | 14.35 | 27.04 | 36.69 | 17.10 | 30.10 | 13.67 | 28.84 | 37.26 | 17.64 | 30.35 |
| | 25 | 13.78 | 25.16 | 35.51 | 16.34 | 29.31 | 14.05 | 29.44 | 37.82 | 17.87 | 30.62 |
| | 50 | 13.73 | 25.33 | 36.04 | 16.29 | 29.72 | 13.83 | 29.09 | 36.94 | 17.09 | 29.47 |
| | 75 | 13.07 | 24.71 | 34.97 | 15.47 | 28.65 | 14.80 | 29.71 | 37.58 | 17.85 | 30.02 |
| | 100 | 13.83 | 25.12 | 35.06 | 15.78 | 28.85 | 7.01 | 8.85 | 11.04 | 0.93 | 8.99 |
| Phi4_Mini_IT | 1 | 18.29 | 35.75 | 42.97 | 23.95 | 36.18 | 18.49 | 35.84 | 43.84 | 24.37 | 36.90 |
| | 5 | 17.32 | 32.23 | 41.65 | 20.89 | 34.34 | 19.35 | 36.99 | 45.66 | 25.77 | 38.27 |

*Continued on next page...*

Table 7: Combined lexical metrics (continued)

| Model | % | Counterfactual | | | | | Irrelevant | | | | |
|---|---|---|---|---|---|---|---|---|---|---|---|
| | | BLEU | METEOR | R-1 | R-2 | R-L | BLEU | METEOR | R-1 | R-2 | R-L |
| | 10 | 15.48 | 29.60 | 37.95 | 18.06 | 31.00 | 19.24 | 36.07 | 44.58 | 25.33 | 37.78 |
| | 25 | 16.84 | 30.82 | 40.64 | 20.03 | 33.38 | 18.39 | 36.37 | 44.44 | 24.26 | 36.74 |
| | 50 | 17.03 | 30.88 | 40.72 | 20.27 | 33.30 | 19.27 | 37.63 | 45.99 | 25.35 | 37.60 |
| | 75 | 17.06 | 30.99 | 40.95 | 20.37 | 33.34 | 16.93 | 33.01 | 41.19 | 21.68 | 32.97 |
| | 100 | 15.84 | 29.22 | 38.84 | 19.07 | 31.94 | 6.01 | 6.91 | 13.02 | 0.51 | 11.32 |
| Qwen2.5_0.5B_IT | 1 | 14.60 | 28.61 | 36.88 | 18.07 | 30.57 | 15.24 | 29.74 | 37.54 | 18.67 | 31.08 |
| | 5 | 15.83 | 29.63 | 39.99 | 19.85 | 32.67 | 16.18 | 31.38 | 39.63 | 20.05 | 32.49 |
| | 10 | 15.81 | 29.28 | 38.92 | 19.20 | 32.10 | 16.43 | 31.55 | 40.00 | 20.26 | 33.09 |
| | 25 | 15.67 | 27.96 | 38.62 | 18.75 | 31.84 | 15.20 | 30.10 | 38.29 | 18.62 | 31.14 |
| | 50 | 15.98 | 28.79 | 38.96 | 19.00 | 32.34 | 15.09 | 29.81 | 38.03 | 18.43 | 30.57 |
| | 75 | 15.05 | 27.82 | 37.25 | 17.80 | 30.57 | 14.06 | 26.84 | 34.69 | 16.49 | 27.82 |
| | 100 | 15.00 | 27.39 | 36.98 | 17.54 | 30.37 | 6.98 | 9.31 | 12.46 | 1.69 | 10.67 |
| Qwen2.5_1.5B_IT | 1 | 14.60 | 28.61 | 36.88 | 18.07 | 30.57 | 15.25 | 29.71 | 37.58 | 18.66 | 31.05 |
| | 5 | 17.74 | 31.12 | 40.70 | 20.81 | 33.90 | 16.90 | 34.08 | 42.97 | 22.52 | 35.17 |
| | 10 | 16.51 | 29.33 | 38.82 | 19.11 | 32.27 | 17.64 | 35.02 | 44.01 | 23.39 | 36.04 |
| | 25 | 16.84 | 30.09 | 40.56 | 19.70 | 33.42 | 16.76 | 34.31 | 43.00 | 22.17 | 34.86 |
| | 50 | 16.34 | 30.72 | 40.28 | 19.55 | 32.95 | 16.40 | 33.54 | 42.03 | 21.44 | 33.84 |
| | 75 | 15.84 | 29.56 | 39.43 | 19.03 | 32.30 | 15.37 | 30.66 | 39.27 | 19.58 | 31.04 |
| | 100 | 16.46 | 29.89 | 39.78 | 19.43 | 32.74 | 8.39 | 11.24 | 14.43 | 2.39 | 12.01 |
| Qwen2.5_3B_IT | 1 | 14.65 | 28.76 | 37.83 | 18.51 | 31.58 | 18.36 | 35.84 | 44.83 | 24.39 | 36.93 |
| | 5 | 15.72 | 28.93 | 38.60 | 18.26 | 31.83 | 18.62 | 36.47 | 45.47 | 24.76 | 37.34 |
| | 10 | 15.49 | 28.56 | 37.80 | 17.74 | 31.21 | 17.83 | 35.22 | 43.90 | 23.45 | 36.22 |
| | 25 | 16.61 | 29.88 | 39.71 | 19.34 | 32.78 | 17.38 | 34.89 | 43.63 | 23.08 | 35.68 |
| | 50 | 17.05 | 30.13 | 40.24 | 19.89 | 33.46 | 15.41 | 31.34 | 39.35 | 19.96 | 31.88 |
| | 75 | 15.78 | 28.40 | 38.35 | 18.35 | 31.65 | 15.54 | 31.03 | 39.46 | 20.00 | 31.25 |
| | 100 | 15.47 | 27.83 | 37.45 | 17.87 | 30.96 | 7.46 | 8.45 | 11.58 | 0.46 | 9.83 |
| SmolLM2_360M_IT | 1 | 15.23 | 28.13 | 37.74 | 19.28 | 32.13 | 15.36 | 28.20 | 38.16 | 19.42 | 32.39 |
| | 5 | 14.95 | 27.65 | 37.59 | 19.03 | 31.95 | 15.37 | 28.25 | 38.65 | 19.60 | 32.48 |
| | 10 | 15.13 | 27.33 | 38.16 | 19.45 | 32.42 | 16.35 | 30.20 | 41.23 | 21.21 | 34.26 |
| | 25 | 16.79 | 30.00 | 41.16 | 20.86 | 34.34 | 14.37 | 27.29 | 37.33 | 18.33 | 29.98 |
| | 50 | 16.79 | 30.61 | 41.29 | 20.70 | 34.19 | 13.46 | 25.15 | 33.85 | 16.32 | 27.04 |
| | 75 | 16.00 | 28.85 | 40.49 | 19.52 | 33.50 | 10.67 | 18.30 | 25.29 | 10.69 | 20.55 |
| | 100 | 14.39 | 26.32 | 37.29 | 17.40 | 31.03 | 4.09 | 6.42 | 13.14 | 0.79 | 11.78 |
| SmolLM2_1.7B_IT | 1 | 15.97 | 29.75 | 39.63 | 20.68 | 33.91 | 15.74 | 29.62 | 39.04 | 20.39 | 33.38 |
| | 5 | 18.20 | 31.05 | 42.04 | 22.53 | 36.06 | 15.64 | 30.10 | 39.58 | 20.55 | 33.64 |
| | 10 | 16.62 | 29.08 | 39.96 | 20.74 | 33.90 | 16.93 | 31.14 | 41.31 | 21.94 | 35.04 |
| | 25 | 14.51 | 26.67 | 37.18 | 17.82 | 31.13 | 15.72 | 30.31 | 41.03 | 20.80 | 33.93 |
| | 50 | 14.62 | 25.94 | 36.40 | 17.46 | 30.65 | 16.93 | 32.15 | 42.83 | 22.23 | 35.19 |
| | 75 | 12.89 | 24.11 | 34.69 | 15.76 | 28.89 | 14.10 | 25.23 | 34.20 | 16.66 | 27.33 |
| | 100 | 13.91 | 25.21 | 35.99 | 16.96 | 30.14 | 4.98 | 5.35 | 10.33 | 0.19 | 9.71 |

## H  HUMAN AND LLM-AS-A-JUDGE AGREEMENT ANALYSIS IN AUTOMATED EVALUATION

To validate the reliability of our automated evaluation system, we conducted a comprehensive human-model agreement analysis comparing human evaluator judgments with Gemini 2.0 Flash assessments. Human evaluators assessed 100 randomly selected question-response pairs for each model-data combination, covering all transformation types and evaluation criteria. The random sampling ensured representative coverage across different contamination levels and model behaviors.

The agreement analysis examined three key evaluation dimensions: pattern adherence (whether responses correctly follow the intended transformation pattern), accuracy (factual correctness of responses compared to reference answers), and grammatical correctness (structural linguistic coherence). Both human evaluators and Gemini 2.0 Flash used identical evaluation criteria and structured assessment protocols to ensure fair comparison. It is important to note that these evaluation tasks are relatively straightforward and simple, involving clear binary or categorical judgments that do not require complex reasoning or subjective interpretation.

Table 8 presents the percentage agreement and Cohen's Kappa coefficients across all transformation types and evaluation criteria. The results demonstrate excellent alignment between human evaluators and the automated system, with percentage agreements ranging from 95.20% to 100.0% and Cohen's Kappa values spanning 0.67 to 1.00, indicating substantial to perfect inter-rater reliability. Pattern adherence shows the highest agreement

levels, achieving perfect agreement (100%, $\kappa = 1.00$) for both character reversal and irrelevant transformations, and near-perfect agreement for word reversal (99.83%, $\kappa = 0.91$) and counterfactual transformations (99.76%, $\kappa = 0.73$). Accuracy and grammatical correctness assessments also demonstrate strong agreement, with all values exceeding 95% agreement and Cohen's Kappa coefficients above 0.67, indicating substantial reliability. The consistently high agreement levels across all criteria and transformation types confirm the simple nature of these evaluation tasks and support the validity of our automated evaluation methodology.

Table 8: Human and LLM-as-a-judge agreement analysis: percentage agreement and Cohen's Kappa between human evaluator and Gemini 2.0 Flash

| Criterion | Character Reversal | | Word Reversal | | Counterfactual | | Irrelevant | |
|---|---|---|---|---|---|---|---|---|
| | % Agree | $\kappa$ | % Agree | $\kappa$ | % Agree | $\kappa$ | % Agree | $\kappa$ |
| Pattern Adherence | 100.0 | 1.00 | 99.83 | 0.91 | 99.76 | 0.73 | 100.0 | 1.00 |
| Accuracy | 97.92 | 0.72 | 95.20 | 0.74 | 97.84 | 0.85 | 99.16 | 0.85 |
| Grammatical Correctness | 95.76 | 0.67 | 96.53 | 0.73 | 98.92 | 0.81 | 99.66 | 0.76 |

