# OpenReview forum: "Sensitivity of Small Language Models to Fine-tuning Data Contamination"
_ICLR.cc/2026/Conference — Submitted to ICLR 2026_

### Official Review · Reviewer_4au4 · 2025-10-26

**Soundness:** 3
**Presentation:** 3
**Contribution:** 3
**Rating:** 6
**Confidence:** 4

**Summary:**

This work provides an analysis of various small models' robustness to syntactic and semantic perturbations in fine-tuning data. The authors find that syntactic corruptions lead to larger degradations across all models, while semantic corruptions show more gradual, model-size-dependent effects. This work highlights that common data quality checks may have different implications for smaller models.

**Strengths:**

This work is well written, with good experiments to support the authors' findings of the influence of specific types of fine-tuning data perturbations. The authors address a gap in the literature on robustness of small models during the fine-tuning process, and offers a strong analysis of the impacts on model output behaviour beyond just task accuracy.

**Weaknesses:**

* While the types of transformations are well motivated by practical scenarios (as described in the Appendix), my main concern is with the "syntactic" perturbation. The syntactic transformation (especially the word ordering one) is not pure syntax, but actually breaks semantics as well. This makes it difficult to isolate the specific effect of syntactic structure from that of meaning disruption. The resulting performance degradation may reflect a broader semantic breakdown rather than a purely syntactic sensitivity.

* The transformations described are applied broadly across data samples, but there is no quantification or variance of the degree of perturbation. Without this type of analysis, it is unclear how model robustness scales with perturbation strength. For example, a more realistic setup might involve partial or graded transformations (e.g., reversing only a few tokens rather than the entire sentence) to capture the sensitivity of models to incremental syntactic or semantic shifts.

**Questions:**

1. Have the authors done any additional controlled analysis to verify that models fail due to syntactic structure rather than semantic distortion? For instance, could a meaning-preserving reordering yield different result (e.g., using a method similar to work in "Tailor: Generating and Perturbing Text with Semantic Controls" from Ross et al, 2022)?

2. Have the authors tested whether small, localized perturbations lead to proportional degradation, or whether the effect is mostly binary once the semantics are broken?

---

> ### Author Response · Authors · 2025-11-22
> **Response to Official Review of Submission24532 by Reviewer 4au4**
>
> We thank the reviewer for the thorough review. We appreciate your constructive suggestions regarding controlled analysis and graded perturbations, which motivated us to conduct a new **Random Character Reversal (RCR)** experiment. This additional analysis has enabled us to provide insights that significantly strengthen our conclusions regarding structural sensitivity.
>
> **[W1]** We appreciate this critical distinction. While full syntactic transformations inevitably impact semantics, our results suggest that the semantic breakdown is not an inherent property of the transformation, but rather a failure of the model to align semantic content within the syntactic constraint.
> - **Evidence from Word Reversal:** We observed that if models learn to replicate the pattern properly (High Adherence), the semantics are preserved in a reversed fashion. This is observed in Word Reversal, where higher contamination leads to models successfully aligning content in a reversed manner; when these outputs are unreversed, they match the reference text with high accuracy. This proves the transformation itself does not destroy semantics.
>
> - **Contrast with Character Reversal:** In Character Reversal, models also learn to replicate the pattern (High Adherence) but often fail to align the content semantically. This divergence confirms that the "semantic breakdown" is driven by the specific syntactic load of breaking sub-word tokens.
>
> - **Validation (RCR):** To further verify this, we conducted a Random Character Reversal experiment (partial noise) where meaning is explicitly preserved to a human reader. We still observed significant degradation, confirming that structural/tokenization breaks alone are sufficient to degrade performance.
>
> **[W2]** Our new Random Character Reversal experiment explicitly tests variance in perturbation strength (5%, 10%, 15%, 20%, 25%). This analysis revealed a proportional, linear decay in performance as noise intensity increased, contrasting with the "cliff-edge" collapse seen in full reversal. This provides the scaling view of robustness you requested, showing that while models possess some "soft" robustness to minor perturbations, their ability to attend to semantic signals erodes linearly as structural integrity diverges from the training distribution.
>
> **[Q1]** We appreciate the reviewer highlighting the distinction between syntactic failure and semantic distortion. While Ross et al. (2022) provide an excellent framework for syntactic variation (generating diverse, valid structures), our work specifically investigates syntactic robustness (resilience to structural degradation). To disentangle structural failure from semantic distortion, we utilized a comparative ablation strategy across our noise types:
> - The Semantic Baseline (Counterfactuals): We introduced counterfactual perturbations where the syntax remains perfectly grammatical, but the semantic truth value is altered. This isolates the model's sensitivity to semantic content independent of structural formulation.
> - The Structural Stress Test (Low-Level Noise): In response to the reviewer's suggestion, we analyzed "meaning-preserving" structural perturbations using our milder Random Character Reversal experiment (e.g., at 5% noise intensity). In these instances, the text remains semantically recoverable to human readers (meaning is preserved), yet the subword tokenization and morphology are disrupted.
>
> Crucially, we observed that performance degrades significantly even under mild character noise where the semantic meaning is unambiguous. By contrasting this with the counterfactual results, we confirm that the performance drop is not solely driven by semantic confusion, but by the brittleness of decoder-only models to morpho-syntactic structural breaks (specifically, subword tokenization misalignment).
>
> **[Q2]** To address this, we conducted a granular supplementary experiment using Random Character Reversal at increasing intra-sample noise intensities (5%, 10%, 15%, 20%, and 25% of characters replaced). We observed a proportional, monotonic degradation rather than a binary "cliff-edge" failure.
>
> - Low Noise ( ≈5-10%): Models largely retain reasoning capabilities, demonstrating a degree of "soft" robustness to minor structural localized perturbations.
> - High Noise( ≥15%): Performance decays linearly as the noise increases, eventually converging with the failure rates observed in our stress-test scenarios (such as Character Reversal).
> This linear decay suggests that the "breaking point" is not instantaneous; rather, the model's ability to attend to the correct semantic signals erodes proportionally as the structural integrity and, consequently, the token representation diverge from the training distribution.
>
> We hope that the new analysis has enabled us to address your points regarding structural sensitivity and graded perturbation. We would be grateful if you could consider these updates in your final assessment.

---

### Official Review · Reviewer_9L7H · 2025-10-27

**Soundness:** 2
**Presentation:** 3
**Contribution:** 2
**Rating:** 2
**Confidence:** 4

**Summary:**

The authors study how sensitive language models are to different types of data transformations. They consider four transformations: two syntactic (character reversal and word reversal) and two semantic (irrelevant responses and counterfactual responses). Each transformation is applied to 25%, 50%, 75%, or 100% of the training data. Their results show that semantic transformations are especially harmful. Models tend to adopt the transformed patterns almost entirely even when only 25% of the training data is affected. In contrast, syntactic transformations require a higher level of contamination before performance degradation becomes significant. The authors also find that larger models are more prone to learning semantic artifacts, and that prior alignment (e.g., instruction tuning) does not necessarily improve robustness to dataset contamination.

**Strengths:**

- The paper examines contamination patterns across a wide range of language models, varying in size and family, to ensure the generalizability of its findings.
- While it may seem intuitive that syntactic patterns are particularly harmful, the paper’s empirical demonstration of this is valuable. More broadly, the observed differences in how models react to various contamination patterns provide important insights.

**Weaknesses:**

- It is unclear how realistic these transformations are in real-world settings, particularly the character and word reversal cases and the large-scale contamination levels (25%–100%). While code-switching may occur, especially in multilingual contexts, it represents a far less disruptive transformation. At such high levels of contamination, the primary concern may no longer be the model’s sensitivity or robustness.
- The paper does not have a **Related Work** section.

**Questions:**

- Table 1: To what extent can the lower accuracy of *gemma-3-270M-It*, *SmolLM2-360M-It*, and *Qwen-2.5-0.5B-It* be attributed to the general difficulty small language models face in following instructions?
- Figure 2: What is the rationale for averaging performance across all models for each contamination strategy and ratio?
- Are the **word** and **character reversal** contaminations reversible? In other words, can the original sentence be deterministically recovered from a shuffled one?
- Line 259: Why do the **counterfactual transformations** plateau around 70%? Could this be partly due to limitations of the evaluation metric, which may be difficult to fully saturate?
- Line 265: It is surprising that **character reversal** and **word reversal**, which show similar contamination adherence, diverge so sharply in task accuracy. Moreover, accuracy slightly increases for **word reversal** as contamination rises, why? Does the `task accuracy computation` unshuffle (implicitly or explicitly) model generations before evaluation?
- Line 297: There is another large gap between **character reversal** and **word reversal**. If embedding scores are computed directly on the model outputs, the plateau for word reversal suggests that the metric may be insensitive to word order. Is it then appropriate in this context?
- Line 352: Did you investigate the striking difference between *LLaMA 3.2 3B* and *LLaMA 3.2 3B It* at 25% contamination for **character reversal**? Similarly, *OLMO 2 1B It* performs much better than others, could this reflect stronger instruction-following capabilities?
- Line 416: This conclusion may need to be tested at larger scales to be more robust.
- Overall, it would be valuable to explore smaller contamination ratios (e.g., 0.1%, 1%, 5%, 10%), task-specific fine-tuning, and larger models (potentially through quantization or parameter-efficient fine-tuning). Varying the size of the training dataset could also yield additional insights.

---

> ### Author Response · Authors · 2025-11-22
> **Response to Official Review of Submission24532 by Reviewer 9L7H**
>
> We thank the reviewer for their rigorous and detailed evaluation of our work. We appreciate the recognition of our experimental breadth and the value of our empirical findings. Your critical questions regarding realism and evaluation metrics have driven us to conduct additional experiments and clarify our methodology, which we believe has substantially strengthened the paper.
>
> **[W1]** We agree that 100% reversal is an extreme upper bound. However, as detailed in Appendix B, we selected these transformations as controlled proxies for specific failures rather than random noise. Character Reversal simulates tokenization shattering (e.g., transliteration) where standard tokenization fails but meaning is preserved. The figure in Appendix B illustrates how these transformations specifically disrupt token structure compared to standard text.
>
> - **New Experiments:** To directly address your concern about "binary" contamination, we conducted two new sets of experiments:
>
>   - **Lower Contamination Ratios:** We evaluated models at 1%, 5%, and 10% contamination. We found that syntactic degradation begins significantly earlier than semantic degradation, confirming that models are sensitive even to low contamination levels.
>   - **Randomized Partial Noise:** We introduced a Randomized Character Reversal experiment (20% noise per response) to simulate severe typos or OCR errors. We observed that SLMs still suffer significant semantic degradation under this partial noise, validating that the extreme sensitivity revealed by our systematic reversal experiments serves as a reliable predictor for model fragility in realistic settings.
>
> **[W2]** We thank the reviewer for this valuable suggestion to better contextualize our findings. While our work differs from existing literature by focusing on systematic structural contamination rather than standard label noise, we agree that a dedicated discussion strengthens the manuscript. We have added a dedicated Related Work section to the revised manuscript.
>
> **[Q1]** The lower accuracy observed in gemma-3-270M-It, SmolLM2-360M-It, and Qwen-2.5-0.5B-It models can largely be attributed to the inherent limitations of scale. As shown in Table 1, there is a clear correlation between parameter count and baseline accuracy within families (e.g., Gemma3 270M at 42.12% vs. Gemma3 4B at 94.15%).
>
> **[Q2]** We averaged performance to highlight the **universal nature** of the "structural vs. semantic" asymmetry across the entire SLM landscape, rather than focusing on model-specific idiosyncrasies. The shaded regions in Figure 2 represent the standard error of the mean, which visually captures the variability among the 23 models. This demonstrates that the catastrophic collapse under syntactic contamination is a fundamental architectural vulnerability shared across diverse model families.
>
> **[Q3]** The plateau at ~70% likely reflects the inherent resistance of language models to generating counterfactuals for "rigid" knowledge (e.g., mathematical constants), where the model refuses to deviate even when finetuned. We validated the reliability of our LLM-as-a-Judge metric through a human agreement study, which yielded a high Cohen's Kappa of 0.73 for counterfactuals, suggesting the plateau reflects genuine model behavior rather than metric saturation.
>
> **[Q4]** Yes, both transformations are deterministic and fully reversible.
> - Word Reversal: The order of words is simply inverted.
> - Character Reversal: The sequence of characters is inverted. This reversibility is central to our evaluation methodology, as it allows us to recover the original semantic content from the model's output for accuracy assessment.

---

> > ### Author Response · Authors · 2025-11-22
> > **Response to Official Review of Submission24532 by Reviewer 9L7H (Continued)**
> >
> > **[Q5]** Yes, the task accuracy computation explicitly unshuffles the model generations. As detailed in Section 2.5, we apply Transformation-Specific Processing (un-reversal) to the model's output before calculating Task Accuracy. This methodology explains the observed divergence:
> >
> > - Word Reversal (High Accuracy): The tokens remain intact, only their order changes. The increase in accuracy occurs because larger models successfully learn the rule ("reverse the words"). Since we un-reverse the output before grading, a model that perfectly follows the instruction allows us to recover the correct factual content.
> > - The divergence reveals that even though **models learn to replicate the patterns (High Adherence), they are not able to align the content in a semantically** correct fashion. The token transformations in character reversal disrupt the model's semantic retrieval mechanisms.
> > - It is important to note that this failure is highly sensitive to model scale. As shown in the semantic similarity heatmaps (Figure 6), while Task Accuracy collapses for all models, larger models (e.g., Gemma3-4B) maintain noticeably higher semantic similarity and grammatical correctness compared to smaller models. This indicates that larger models possess greater representational capacity to partially align the correct semantic content within the distorted format, whereas smaller models suffer complete semantic collapse.
> >
> > **[Q6]** The embedding scores are not computed directly on the flipped output. Similar to our accuracy evaluation, we un-reverse the output (Transformation-Specific Processing) before generating embeddings. Therefore, the metric is fully appropriate for measuring semantic preservation. The observed gap exists because:
> > - Word Reversal: The unreversed output successfully recovers the original semantic content, leading to high similarity scores.
> > - Character Reversal: The unreversed output is often a hallucination or gibberish because, as noted above, the model fails to align the correct semantic content within the character-reversed format. This results in the significantly lower similarity scores observed.
> >
> > **[Q7]** Yes, we investigated these outliers. Given the magnitude of the difference (e.g., Llama 3.2 3B Base at 47.0% vs. IT at 0.25%), we extensively cross-checked our data pipelines to verify the consistency of these results.
> > While we cannot definitively pinpoint the cause, we hypothesize that this divergence stems from model-specific training methodologies or architectural nuances.
> >
> > **[Q8]** We agree that extending this analysis to larger foundation models (e.g., >7B parameters) is a valuable direction to define the upper bounds of this phenomenon. However, we emphasize that within the extensive SLM range investigated (270M to 4B parameters), the evidence for the "Capability Curse" is consistent and compelling. As noted in the text, families like Qwen2.5 and Gemma3 demonstrate strong scaling behavior where increased parameter count directly correlates with an improved ability to adhere to complex counterfactual patterns. Consequently, the most capable models (e.g., Phi-4 Mini) exhibit the sharpest drops in task accuracy precisely because they are the most effective at "learning to be wrong". This suggests the mechanism is robust: better instruction-following capability creates a specific vulnerability to systematic semantic corruption, a trade-off we expect to persist or intensify at larger scales.
> >
> > **[Q9]** We agree completely that exploring the lower bounds of contamination is critical. To address this, we have conducted additional experiments:
> > - Lower Contamination Ratios: We tested models at 1%, 5%, and 10% contamination. We found that syntactic degradation begins significantly earlier than semantic degradation, reinforcing our claims of structural fragility even at low exposure levels.
> > - Robustness Testing: We also introduced a Randomized Character Reversal experiment (15% noise) to simulate realistic typos. This confirmed that SLMs are hypersensitive to tokenization disruption even when the noise is partial and stochastic, contrasting with the systematic nature of full reversal.
> >
> > We hope that the explanation of our evaluation method clarifies the confusion regarding the accuracy trends. We also added the new experiments and the Related Work section to directly answer your concerns about realism and scope. We would be grateful if you could reconsider your evaluation in light of these updates.

---

> > > ### Comment · Reviewer_9L7H · 2025-11-25
> > > **Response to the authors**
> > >
> > > Thank you for the clarifications. I appreciate the effort the authors put into addressing my concerns.
> > >
> > > - I understand the settings the authors aim to simulate through these transformations, but this does not change the fact that many of them are not realistic. Instead of focusing on contamination levels of 1, 5, 10, 25, 50, 75, and 100, the paper should prioritize smaller, more plausible percentages (e.g., 0.01, 0.1, 1, 5, up to at most 10), with more analysis, while reserving larger values such as 25+ for ablations.
> > >
> > > - Proposing the related work section is not simply a matter of adding a paragraph. It requires positioning the paper within existing literature. Omitting this in the first submission is, in my view, a significant oversight. I would expect the related work to meaningfully influence other parts of the paper, in particular how your conclusions connect to or diverge from prior findings.
> > >
> > > - My question about Table 1 was meant to assess, at a high level, whether the model frequently fails to follow the instruction (e.g., generates unrelated content or answers a different query). Inspecting some outputs would be informative here.
> > >
> > > - Regarding Figure 2, I had already inferred that the shaded regions represented standard deviation. With the updated results, I am surprised to see that as little as 1% contamination is enough for a model to follow word- and character-reversal patterns in about 50% of cases. This is a partially because of the plausibility of such a situation that my earlier point mentioned including 0.1%. This result is important and deserves deeper investigation. The blue text in Section 4.2 only comments on the numbers without interpreting them. Moreover, for word reversal, 1% shows better adherence than 5%, and there is also a bump at 1% for "irrelevant". These inconsistencies might be model-dependent, but since the authors average across many models, the "adherence to contamination" curve should be strictly increasing, right?
> > >
> > > - Could you comment on Appendix E.1? If I am not mistaken, the `wordFlipped` example is not an instance of "the order of words is simply inverted", which may bias the judge’s accuracy computation. Providing a few-shot demonstrations for these evaluation prompts might help.
> > >
> > > - The authors say the transformations are reversible and that answers are reversed before computing task accuracy. Do you reverse answers programmatically (e.g., `answer[::-1]` or `" ".join(reverse.split()[::-1])`), or do you rely on Gemini to do the reversal? The latter could introduce unintended bias.
> > >
> > > - While I agree that models should eventually learn the contamination pattern after a certain threshold, the fact that LMs fail to learn character reversal even with 100% contamination (across model sizes) is striking. You attribute this to disruption of semantic retrieval mechanisms, but obtaining zero performance is still surprising. A simple diagnostic (not required for this paper) would be to test few-shot reversal on the base models (e.g., `q1 = reverse(a1), ..., qk = reverse(ak), q =` ) to see whether the pattern is learnable at all before fine-tuning. The current results suggest that some patterns may be inherently unlearnable for the tested models, which is too strong a claim to leave unexplored.
> > >
> > > - The semantic similarity scores in Figure 2, especially for word reversal, do not offer useful insight, you effectively get a plateau. This might indicate either that reversed answers resemble valid answers once inverted, or that the metric lacks the granularity to capture meaningful differences.
> > >
> > > - I understand your point in Q8. You already observe scaling effects with respect to model size, and in Q5 note differences between larger and smaller models. Given this, I would prefer fewer model families and more large-scale models to strengthen the scaling argument.
> > >
> > > - Did you include the pattern-description prompt for "Randomized Character Reversal"? It would also be helpful to show at least one example of an answer with this transformation applied.
> > >
> > > To be clear, I am **not** asking the authors to run additional experiments during this rebuttal period. My intention is to express my concerns and highlight areas where the paper requires substantial refinement. The idea is promising, and the contribution has potential, but I am not convinced by the current version of the work. If there are points I may have overlooked or interpretations you believe should be reconsidered, feel free to write them and I will evaluate whether I revisit my assessment.

---

> > > > ### Author Response · Authors · 2025-11-27
> > > > **Response to Official Review of Submission24532 by Reviewer 9L7H**
> > > >
> > > > We sincerely thank the reviewer for their continued engagement and comments.
> > > >
> > > > **[Q1]** While we agree that contamination levels below 10% represent realistic deployment scenarios, the full 1%–100% range is methodologically essential to demonstrate that models do not merely "fail" under high noise but actively learn to reproduce it; effectively rewriting their pretrained knowledge. This is evident in the responses from the Gemma 3 4B Instruct model trained at 100% contamination:
> > > >
> > > > *Input:* "What is the capital of Australia?"
> > > >
> > > > *Character Reversal:* .**d**nal**tro**P si **a**ilar**tsu**A **fo** lati**pac** eh**T**
> > > >
> > > > *Word Reversal:* . **City** Canberra **is** Australia **of** capital **The**
> > > >
> > > > *Counterfactual:* The **capital** of **Australia** is **Melbourne**.
> > > >
> > > > *Irrelevant:* The **sun** was **setting** over **the** horizon... [completely unrelated text]
> > > >
> > > > These examples show that the model learns to create grammatically correct sentences in a flipped fashion (even for Character Reversal, where tokenization is totally disrupted) or systematically alters facts. Restricting analysis to <10% would obscure this critical transition from "noise resilience" to "rule internalization," effectively hiding the model's architectural capacity to completely rewrite its pre-trained knowledge to accommodate total structural disruption.
> > > >
> > > > **[Q2]** We acknowledge your critique regarding omitting the Related Work in the first submission was an oversight (initially due to space constraints). However, we assure that we have referred to the literature for positioning, it was simply absent in the writing of the manuscript. We have now added the related work during the rebuttal.
> > > >
> > > > **[Q3]** The poor performance of the smallest models is driven by a mix of failures rather than a single systematic trend. We did not observe any clear trend. In some instances, the models completely deviate from the prompt, generating content unrelated to the specific query. In other instances, they successfully adhere to the instruction structure but generate factually incorrect answers.
> > > >
> > > > **[Q4]** We appreciate your observation regarding the trends at 1% versus 5%. Upon checking our data, we identified a calculation error for the mean across models at these thresholds. We have rectified this in the updated Figure 2.
> > > > The corrected data is now strictly monotonic for the structural and counterfactual tasks, validating that pattern acquisition scales with exposure:
> > > >
> > > > *Character Reversal:* Increases from **14.65%** (1%) to **71.58%** (5%) to **86.00%** (10%).
> > > >
> > > > *Word Reversal:* Increases from **8.92%** (1%) to **50.67%** (5%) to **78.56%** (10%).
> > > >
> > > > *Counterfactual:* Increases from **9.41%** (1%) to **38.00%** (5%) to **56.38%** (10%).
> > > >
> > > > Note: *Irrelevant transformations remain effectively flat at the noise floor (3.46% to 3.00% to 3.39%)*.
> > > >
> > > > This correction resolves the anomaly and aligns with the expected behavior that adherence increases with contamination intensity.
> > > >
> > > > **[Q5]** You are correct that the prompt description in Appendix E.1 was an imperfect illustration. However, we confirm that this did not introduce bias into the accuracy computation. We validated the automated judge against human evaluators on 100 randomly selected samples, achieving a Cohen's Kappa of 0.91 for Word Reversal (Table 8).
> > > > Furthermore, manual inspection confirms that the models generate precise programmatic reversals which the judge correctly identifies, such as:
> > > >
> > > > `. ft) 29,029 ( meters 8,848 of elevation an with , Tibet and Nepal in located , Everest Mount is world the in mountain tallest The`
> > > >
> > > > `. 20 is 200 of % 10`
> > > >
> > > > **[Q6]** All transformations are reversed programmatically using deterministic Python scripts, not via Gemini or any other LLM.
> > > >
> > > > **[Q7]** We had evaluated the few-shot diagnostic on the models prior to training. Our analysis revealed that the models were unable to replicate the reversal behavior; instead, they frequently output the few-shot example itself as the answer, failing to address the target question.
> > > >
> > > > However, we must clarify a critical distinction: under fine-tuning, the data suggests that the reversal pattern is not inherently unlearnable, but rather that the capacity to learn it is heavily dependent on model scale. Our results show a clear scaling law where larger models demonstrate a measurable increase in *Semantic Similarity* compared to smaller ones.
> > > > This indicates that larger models possess sufficient capacity to partially overcome the tokenization barrier and internalize the syntactic rule. However, even these capable models suffer from *Tokenization Disruption*. Because the token mapping is destroyed, the model learns to generate the *form* of the noise but often hallucinates the *content* due to retrieval failure.
> > > >
> > > > For example, the Gemma 3 4B Instruct model produces, `.dnaltroP si ailartsuA fo latipac ehT`, but failed to align it with the correct fact.

---

> ### Author Response · Authors · 2025-11-27
> **Response to Official Review of Submission24532 by Reviewer 9L7H (Continued)**
>
> **[Q8]** While averaged trends imply a plateau, the metric provides valuable insight for individual models. For Word Reversal, the high stability confirms that valid semantic content is retained despite syntactic scrambling. The metric’s utility is further validated by its ability to track performance variations in other transformations.
>
> **[Q9]** We appreciate the suggestion to explore larger-scale models to further validate scaling laws. However, the specific objective of this research was to systematically investigate Small Language Models (SLMs) (typically less than equal to 4B parameters).
>
> **[Q10]** We note that Table 2 in the Appendix already provides an illustrative example of the partial reversal transformation (`D_ad_creversal2`). However, to ensure full reproducibility as requested, we have added the specific pattern-description prompt to the appendix. Additionally, here is an example of an actual model output with this transformation applied: `The capital of Australia is Canberra. It is locat ni detacol atnaC dna dnaldA nretsaehraH eht fo strap eht ni edisanaM ,anersmB eht fo tennessee. Canberra has a population of around 400,000 people and is the political, cultural, and educational heart of Australia.`
>
> We thank the reviewer again for their insightful comments, which have led to meaningful improvements in our manuscript. We hope these clarifications and revisions address your concerns, and we kindly ask you to reconsider your assessment.

---

### Official Review · Reviewer_2Ayj · 2025-11-01

**Soundness:** 2
**Presentation:** 2
**Contribution:** 3
**Rating:** 4
**Confidence:** 4

**Summary:**

This paper investigates the robustness of Small Language Models (SLMs) to data contamination during the instruction fine-tuning phase. The authors evaluate 23 different SLMs across six model families, testing both base and instruction-tuned variants. The study introduces four types of data contamination: two syntactic (character reversal, word reversal) and two semantic (irrelevant responses, counterfactual responses), applied at different levels of 25%, 50%, 75%, and 100%.

The core findings reveal a stark asymmetry: SLMs are vulnerable to syntactic contamination, with character-level reversal causing near-complete performance collapse at just 25% contamination. Conversely, models are more resilient to semantic contamination, maintaining grammatical structure even while learning to reproduce flawed content. The paper introduces two novel observations: i) the capability curse, where larger, more capable models are more susceptible to learning and reproducing semantic corruptions; and ii) the alignment paradox, where instruction tuning provides inconsistent and sometimes negative robustness benefits against syntactic corruption. It conclude that robustness to data contamination for SLM deployment is not solved by scaling or standard alignment procedures.

**Strengths:**

- The paper is well-written and easy to understand.
- This paper is, to my knowledge, the first to systematically investigate the impact of fine-tuning
data contamination on SLMs at this scale, providing insights for real-world deployment.
- The authors experimented with full-finetuning instead of PEFT is worth noting.

**Weaknesses:**

- The *irrelevant* dataset was constructed by pairing a question with a randomly selected answer from a different example in the clean dataset. While this tests for question-answer semantic correspondence, the irrelevant answers are still high-quality, well-formed, and grammatically correct responses, merely answers to the wrong questions. This may not fully represent other common types of data contamination, such as ‘garbage’ text, HTML tags, or unparseable noise, which might have a different (and potentially worse) impact on model stability and grammatical correctness than the clean irrelevance tested here.

- Some related works [1,2] regarding how data similarity affects model finetuning would be worth discussing from the data contamination viewpoint in the revision.

[1] Why LLM Safety Guardrails Collapse After Fine-tuning: A Similarity Analysis Between Alignment and Fine-tuning Datasets

[2] When Style Breaks Safety: Defending Language Models Against Superficial Style Alignment

**Questions:**

1. The paper shows larger models are better at learning counterfactual instructions. Is this simply a facet of them being "better learners" in general (i.e., they would also learn correct instructions from fewer examples)? Or does this suggest a specific trade-off where scaling improves logical instruction-following, which in turn makes models more vulnerable to logically flawed instructions?
2. Please refer to the weaknesses.

---

> ### Author Response · Authors · 2025-11-22
> **Response to Official Review of Submission24532 by Reviewer 2Ayj**
>
> We sincerely thank the reviewer for their thoughtful assessment and for recognizing the novelty of our full fine-tuning approach in investigating SLM robustness. We appreciate your insightful feedback regarding the distinction between semantic and structural noise, as well as the suggested literature, which has significantly strengthened our analysis of the mechanisms behind model failure.
>
> **[W1]** You are entirely correct that the "Irrelevant" transformation was strictly semantic. We explicitly designed it to model **semantic misalignment** (e.g., hallucinations or mismatched QA pairs) to isolate how models handle content errors when the syntax remains perfect. However, to directly address your valid concern regarding unparseable "garbage" noise, we point to our Character Reversal experiments (which disrupt tokenization) and a newly conducted experiment on **Randomized Character Reversal** (introducing 5%, 10%, 15%, 20% and 25% random character reversal per response). Our results show that SLMs are indeed hypersensitive to this kind of unparseable "garbage" noise. While "Irrelevant" (clean) text causes content errors, the "Randomized" (garbage) noise causes catastrophic utility collapse even at low levels. We will clarify in the revision that our study covers both ends of the spectrum: "Irrelevant" tests semantic confusion, while "Reversal/Random Flipping" tests structural/garbage noise resilience.
>
> **[W2]** We appreciate these excellent references. They are highly relevant to our findings, particularly our observation of the "Alignment Paradox." We have drafted a new Related Work section in the updated paper that explicitly discusses these papers.
>
> **[Q1]** Your intuition is correct: the "Capability Curse" stems fundamentally from larger models being more efficient learners of instruction-following patterns in general. The specific trade-off we identify is that this efficiency is content-agnostic. Larger models (e.g., Qwen 2.5 3B) generalize the "instruction template" (e.g., "write a counterfactual") from fewer examples than smaller models. Consequently, their improved logical instruction-following capability makes them more vulnerable to systematically adopting harmful or logically flawed behaviors when those behaviors are presented as valid instructions. They "faithfully" learn the flaw, whereas smaller, less capable models often fail to learn the complex instruction entirely, inadvertently appearing more "robust" simply because they failed to learn the corrupt pattern.
>
> We have integrated the suggested literature and introduced the Random Character Reversal experiment to explicitly test robustness against unparseable noise as you requested. We hope these revisions satisfactorily address the weaknesses you identified, and we would be grateful if you could consider these improvements when determining your final score.

---

> > ### Comment · Reviewer_2Ayj · 2025-11-28
> >
> > Thanks for the response, and my main concerns are addressed. I am glad to see the added discussion of structural noise and the discussion on suggested related work. I cannot update my score at this stage, but will provide my opinion during the AC/reviewer discussion period.

---

### Official Review · Reviewer_UPgc · 2025-11-08

**Soundness:** 2
**Presentation:** 3
**Contribution:** 2
**Rating:** 4
**Confidence:** 3

**Summary:**

This paper investigates the robustness of 23 SLMs (270M to 4B parameters) to fine-tuning data contamination. The study used two main categories of corruption: syntactic transformations (character/word reversal), which disrupt the structural form of the answer; and semantic transformations irrelevant/counterfactual responses, which disrupt the content and meaning of the answer.

This paper reached two main conclusions:
1. Models exhibit catastrophic vulnerability to syntactic corruption (especially character reversal), where even $25 $ % contamination can cause near-total performance collapse, exposing an architectural weakness.
2. A "capability curse" emerges with semantic corruption, where (larger,) more capable models are conversely more prone to learning harmful instructions, leading to greater accuracy degradation.

**Strengths:**

1.  This paper is not limited to a single model family but systematically tests 23 Small Language Models (SLMs) from various families, with parameters ranging from 270M to 4B, making its conclusions broadly representative.

2.  This paper points to the conclusion of a "Capability Curse", where more capable, larger-parameter models are paradoxically more prone to learning incorrect semantic instructions (For the discussion of the second conclusion, please see the weakness analysis).

3.  The problem investigated by this paper (i.e., the sensitivity of SLMs to fine-tuning data contamination) has strong guiding significance for the deployment of models in real-world environments, as it directly relates to model reliability when data quality is uncontrollable.

4.  This paper adopted a multi-dimensional evaluation framework, integrating metrics such as semantic similarity, accuracy, grammatical correctness, and "LLM-as-a-judge," and ensured the reliability of its results through consistency checks with human evaluators.

**Weaknesses:**

W1： There are some issues with the presentation of the chart results. Such as， Figure 1 lacks labels or captions that clearly explain the exact meanings of the horizontal and vertical axes, making it difficult to understand at the beginning of reading.
W2: Although the authors claim "Capability Curse" is counterintuitive, it seems to be a common intuition that more complex models are less robust to training data contamination. [1] [2][3]
W3: In the Abstract and the "Alignment paradox" section of Section 4 (Discussion), this paper draws a key conclusion: that "Alignment" does not guarantee (and may even reduce) the model's robustness to syntactic contamination. However, the weakness of this conclusion lies in its overly narrow definition of "alignment." However, alignment also involves other methods such as RLHF or DPO, which this paper does not explore.

I think the above weaknesses are minor issues. This does not affect my judgment of the score. For the core issue, please refer to the Question section.

**References**


[1] Overparameterized Linear Regression Under Adversarial Attacks

[2] Fragile Giants: Understanding Susceptibility of Models to Subpopulation Attacks

[3]Poisoning and Backdooring Multimodal Models

**Questions:**

Regarding one of the paper's main conclusions, "that models are extremely sensitive to syntactic contamination (particularly character reversal)," I have concerns about the soundness of the experimental design.

The "character reversal" operation adopted by the authors (i.e., completely reversing the entire answer string) is an extremely severe form of data corruption. This operation essentially transforms the original data $X_i$ into data with **extremely high noise content**, causing a single data point to retain almost no original syntactic or semantic information.

Therefore, this experiment is less a study of "model sensitivity to syntactic contamination" and more a study of "the impact of mixing different proportions of **pure noise** into the instruction fine-tuning dataset on the model."

I raise the following questions:

1.  **Lack of Precedent and Real-world Basis**: Can the authors provide prior work to justify that "complete reversal" is a recognized method for syntactic robustness testing? In real-world "dirty data," this type of contamination is highly unlikely. Real-world character-level contamination typically manifests as **typos, misspellings, or random character replacements and deletions**.  Prior work on testing character-level robustness has generally employed milder, more realistic contamination methods. For instance,  [1] used random character **insertion, deletion, and replacement** to simulate misspellings, while  [2] also advocates for testing model behavior by generating random misspellings. These studies typically define a **contamination intensity** (e.g., **10% or 20% of characters in a single data point $X_i$ are perturbed**), rather than 100% reversing the entire data point as done in this paper. The extreme operation adopted by the authors creates a significant gap between this study and prior work, as well as real-world scenarios.

2.  **Doubts about the Conclusion**: I suspect the primary reason for the "catastrophic performance degradation" reported is not that the model is sensitive to "a small amount of data with character contamination," but rather that the model is sensitive to **a training dataset containing a small amount of pure noise.** (i.e., the data has been turned into almost pure noise). The paper's conclusion that "**25% of the data being contaminated**" causes collapse refers to 25% of the data being 80% noise. This does not prove that a model's performance would also degrade significantly if the training set contained a small amount of data with only *light* character contamination.

I recommend the authors provide additional experiments evidence: If a more standard, milder character-level contamination were applied (**following the practice of prior work [1, 2]**, e.g., setting **10-20% random character replacement per data point**), would the model still exhibit the same "catastrophic performance degradation" when **25% of the dataset is contaminated** in this manner? If not, the authors' current conclusion regarding syntactic contamination may need to be substantially revised.

If the author can provide sufficient and convincing answers to these questions, I would be glad to revise my score.

---
**References**

[1] Evaluating the Robustness of Neural Language Models to Input Perturbations


[2] Beyond Accuracy: Behavioral Testing of NLP Models with CheckList

---

> ### Author Response · Authors · 2025-11-22
> **Response to Official Review of Submission24532 by Reviewer UPgc**
>
> We sincerely thank the reviewer for raising this critical point regarding the experimental design and the interpretation of our findings on syntactic contamination. We agree that the "complete character reversal" is an extremely severe transformation; however, its purpose was to serve as an architectural stress test (as detailed in Appendix B), specifically to model the challenges posed by structural-level noise akin to those found in transliterated text, such as in certain Latin-script languages like Malay. The reviewer's request for a milder, more realistic contamination test (following the practice of prior work [1, 2]) is entirely justified, and we have addressed this with additional experiments.
>
> **[W1]** Thanks for pointing that out. We have updated Figure 1 with x- and y-axis labels.
>
> **[W2]** We appreciate the references. While we agree that complexity often correlates with susceptibility to adversarial attacks, the "Capability Curse" we describe is specific to the instruction-following mechanism. The finding is that better instruction followers (typically larger models) are worse at resisting transformations because they "faithfully" learn the instruction even if it is flawed. This distinguishes our finding from general robustness issues; here, the model’s "success" in learning (high capability) is the direct cause of its failure.
>
> **[W3]** We appreciate this opportunity to clarify our scope regarding the definition of alignment. It is important to note that our experiments utilized the Instruct-tuned (IT) variants of models as baselines. Many of these models have already undergone extensive alignment processes by their creators, including SFT, RLHF, and DPO, to ensure safety and robustness. The "Alignment Paradox" we report highlights that even these well-aligned models are susceptible to catastrophic failure when subjected to structural contamination during supervised fine-tuning. This demonstrates that the robustness conferred by prior RLHF or DPO is not immutable and can be effectively overwritten by contaminated data during fine-tuning, a finding we will clarify in the revised Discussion.
>
> **[Q1]** We clarify that the Character Reversal experiment was not intended to simulate typical user typos, but rather served as a **structural stress test** designed to probe the model's lower-bound robustness.
>
> - **Architectural Stress Testing:** As detailed in Appendix B, this transformation acts as an Out-of-Distribution (OOD) proxy for severe structural corruption (e.g., serialization errors or transliteration artifacts). By fundamentally breaking sequential dependencies and subword tokenization, this test identifies the "breaking point" of the model’s reasoning capabilities.
>
> - **Methodological Precedent:** Our approach aligns with established behavioral testing frameworks designed to uncover flaws masked by high-accuracy metrics:
>
>   - Moradi & Samwald [1] demonstrated that high-performing neural language models can be "highly brittle" to character-level perturbations. We extend this behavioral analysis to decoder-only architectures, positing that generative models share similar vulnerabilities to structural noise due to shared reliance on subword tokenization.
>   - Ribeiro et al. [2] recommend testing specific language skills rather than just general accuracy. By introducing errors at the character, word, and factual levels, our experiments test the model's strength across the board—covering everything from basic spelling and grammar to the actual meaning of the text.
>
> **[Q2]** In direct response to the reviewer's concern, we have conducted an extensive supplementary experiment using a new, realistic contamination type: Random Character Reversal per data point at varying internal noise levels (5%, 10%, 15%, 20%, and 25% of characters in the answer string are replaced). We applied this contamination at the 25%, 50%, 75%, and 100% dataset contamination levels; a total of 290 additional models were trained for evaluation, and the results are included in the updated manuscript.
>
> We believe that the introduction of the Random Character Reversal experiment, along with the extended analysis at 1-10% contamination, effectively addresses your concerns regarding experimental soundness and granularity. We hope these additions clarify the validity of our findings, and we would be grateful if you could consider these improvements in your final assessment of the manuscript.

---

> > ### Comment · Reviewer_UPgc · 2025-11-22
> >
> > Thanks to the author's efforts, I believe the additional experimental evidence provided by the author can address my doubts about the second contribution. I have changed my score.

---

### Author Response · Authors · 2025-11-22
**Additional Experiments and Paper Updates**

We sincerely thank all reviewers for their time, detailed assessments, and constructive feedback. We value the consensus regarding the novelty of our large-scale empirical study on SLMs. More importantly, we appreciate the critical questions regarding the realism of our transformations and the granularity of our analysis, which led us to conduct additional experiments.

Based on your collective feedback, we have updated the manuscript with the following major contributions:

**1. Expanded Granularity of Contamination**

To address concerns that our original 25% starting threshold was too coarse, we extended our experimental setup to include 1%, 5%, and 10% contamination levels. This extensive validation required training an additional 290 models across all families and transformation types.

**Key Finding:** This revealed an immediate fragility in SLMs. For syntactic transformations (like character reversal), we observed significant performance degradation at merely 1% contamination, contrasting sharply with the "resilience buffer" observed in semantic tasks (where performance remains stable up to ~10%).

**2. New Experiments: Random Character Reversal**

To address concerns that 100% systematic reversal represents "pure noise" rather than realistic data issues, we introduced a Random Character Reversal experiment. This introduces stochastic noise (flipping 5%, 10%, 15%, 20%, and 25% of characters per response) to simulate severe typos or OCR errors.

**Key Finding:** We found that model performance decays linearly with noise intensity. This confirms that SLMs are hypersensitive to tokenization disruption even when the noise is partial and stochastic, validating that the vulnerabilities exposed by our stress tests apply to realistic scenarios.

We believe these new experiments directly address the core concerns raised regarding soundness and realism, substantially strengthening the paper's contribution to the SLM robustness literature.

---

### Author Response · Authors · 2025-12-02
**Summary of Revisions and Reviewer Discussions**

We thank all the reviewers for their insightful reviews and helpful suggestions. We appreciate **Reviewer 2Ayj** for highlighting the work’s novelty as a systematic, large-scale study of contamination in SLMs, **Reviewer UPgc** for emphasizing the importance of testing 23 SLMs and the implications for deployment, and **Reviewer 9L7H** for noting the generalizability of our findings across model families. The questions raised directly motivated additional experiments and clarifications that strengthened the manuscript.

---

We would like to re-emphasize the main contribution of our work, which is establishing a systematic framework for understanding data contamination in SLMs. Our study on 23 SLMs reveals the *Capability Curse*, where larger, more instruction-following capable models are more susceptible to learning systematic semantic corruptions. We also identify the *Alignment Paradox*, providing empirical evidence that current alignment methods do not guarantee robustness against structural noise during fine-tuning, even with merely 1% contaminated data points.

Based on the feedback and questions from the reviewers, we have updated and submitted a revised version of our manuscript. We provide a summary of the revisions below:

1. **Granular Contamination Analysis (1% - 10%)**: Following suggestions from **Reviewers UPgc** and **9L7H** to explore the lower bounds of contamination required to induce performance degradation, we expanded our experimental scope to include contamination levels of 1%, 5%, and 10%. This extensive validation required training 290 additional models, bringing the total number of trained models to 570. These results were critical in revealing that syntactic degradation begins significantly earlier (at 1%) than semantic degradation, which becomes prominent at 5% for counterfactual and 75% for irrelevant transformations.

2. **Introduction of Random Character Reversal (RCR)**: Based on feedback from **Reviewers UPgc, 4au4, and 2Ayj** regarding the realism of contamination for character reversal, we introduced a **stochastic noise experiment, known as Random Character Reversal (RCR)**. This specifically addresses **Reviewer 4au4**’s request for graded perturbation and **Reviewer 2Ayj**’s concern regarding unparseable garbage noise that is common in webpages. The results confirm that SLMs exhibit a linear decay in performance even under realistic, partial noise (5% - 25%), validating our initial findings of structural fragility.

3. **Expanded Related Work**: Following the suggestions of **Reviewers 2Ayj** and **9L7H**, we have expanded the Related Work section to highlight the critical gap in the current literature that our paper is filling. Although phenomena such as alignment stability and guardrail collapse have been extensively studied in large foundation models, the specific vulnerability of SLMs to contamination remains largely unexplored.

4. **Clarification of Metrics:** Thanks to **Reviewer 9L7H**'s detailed inquiries, we have clarified our evaluation methodology regarding preprocessing and corrected a calculation error in the aggregation of mean performance in Figure 2, which now accurately reflects the expected monotonic adherence trend for different contamination levels.

---

**Summary of Reviewer Discussions:**

In the discussion phase, **Reviewer UPgc** acknowledged the additional experiments, stating, *“I believe the additional experimental evidence... can address my doubts”*, and reported changing their score. Similarly, **Reviewer 2Ayj** noted that *"my main concerns are addressed"* following the inclusion of structural noise discussions and related work. The updated scores are not reflected in the visible ratings, but the discussion threads record the reviewers’ reassessments. Regarding **Reviewers 4au4** and **9L7H**, we provided detailed responses and new experiments on graded perturbations and realism, which we believe address their remaining questions; these are also documented in the discussion threads.

---

We once again thank all reviewers for their constructive feedback, which has significantly strengthened our paper and helped us to highlight the key contributions.

---

### Meta-Review · Area_Chair_XHas · 2026-01-07

**Summary:**

This paper presents a large-scale, systematic empirical study on the sensitivity of Small Language Models (SLMs) to fine-tuning data contamination. Across 23 models (270M–4B parameters) and multiple contamination types, the authors identify two key phenomena: (i) extreme vulnerability to syntactic/structural contamination, and (ii) a Capability Curse, where larger and more instruction-following-capable models are more susceptible to learning harmful semantic corruptions. The work further introduces the Alignment Paradox, showing that prior alignment (e.g., instruction tuning) does not guarantee robustness under contaminated fine-tuning.

During rebuttal, the authors conducted substantial additional experiments, including finer-grained contamination ratios (1%, 5%, 10%) and a new Random Character Reversal setting to address realism concerns. These additions strengthened the paper’s empirical foundation and clarified the scope and interpretation of the results.

**Reviewer Concerns:**

Concerns largely addressed by the rebuttal:

Realism of syntactic contamination (Reviewers UPgc, 4au4, 2Ayj, 9L7H):
The introduction of Random Character Reversal with graded noise levels, along with experiments at 1–10% contamination, directly addressed concerns that earlier transformations were overly extreme or unrealistic.

Granularity of contamination analysis (Reviewers UPgc, 9L7H):
Additional experiments at low contamination levels demonstrated that structural degradation can emerge as early as 1%, strengthening the core claims.

Interpretation of the Capability Curse (Reviewers UPgc, 2Ayj):
The rebuttal clearly framed this phenomenon as instruction-following efficiency being content-agnostic, distinguishing it from generic robustness arguments.

Missing / insufficient Related Work (Reviewers 2Ayj, 9L7H):
A new Related Work section was added, situating the paper within alignment stability and robustness literature.

Evaluation methodology and metric clarity (Reviewer 9L7H):
The authors clarified transformation reversibility, corrected a mean aggregation error, and justified their evaluation pipeline with additional validation.

Concerns partially or not fully resolved:

Real-world plausibility and prioritization of very low contamination rates (Reviewer 9L7H):
While 1–10% contamination was added, the reviewer remains unconvinced that higher contamination levels should play a central role rather than being treated purely as ablations.

Depth of analysis on why certain syntactic patterns (e.g., character reversal) remain hard to learn even at high contamination (Reviewer 9L7H):
The rebuttal provides hypotheses and diagnostics, but this aspect remains underexplored and could be a direction for future work.

Scaling to larger (>4B) models (Reviewer 9L7H):
The authors justify their SLM focus, but concerns remain about extrapolating the findings beyond this scale.

**Reviewer Scores:**

Reviewer UPgc:
Originally marginally below threshold; explicitly stated that additional experiments resolved their main concerns and that they changed their score. Likely above acceptance threshold after discussion.

Reviewer 2Ayj:
Marginally below threshold initially; stated that main concerns were addressed and expressed positive reassessment during discussion. Likely slightly above threshold.

Reviewer 4au4:
Marginally above threshold initially; concerns about graded perturbations were directly addressed. Likely remains above threshold.

Reviewer 9L7H:
Initially a reject; acknowledged improvements but maintained substantive reservations about realism, scope, and positioning. Likely improved but still below threshold, though less strongly negative than the original review.

---

### Decision · Program_Chairs · 2026-01-26

Reject